# Modular, robust, and extendible multicellular circuit design in yeast

**Alberto Carignano[1], Dai Hua Chen[1], Cannon Mallory[1], R Clay Wright[2], Georg Seelig[1,3]\*, Eric Klavins[1]\***

[1]Department of Electrical and Computer Engineering, University of Washington, Seattle, United States; [2]Department of Biological Systems Engineering, Virginia Tech, Blacksburg, United States; [3]Paul G Allen School of Computer Science and Engineering, University of Washington, Seattle, United States

**Abstract** Division of labor between cells is ubiquitous in biology but the use of multicellular consortia for engineering applications is only beginning to be explored. A significant advantage of multicellular circuits is their potential to be modular with respect to composition but this claim has not yet been extensively tested using experiments and quantitative modeling. Here, we construct a library of 24 yeast strains capable of sending, receiving or responding to three molecular signals, characterize them experimentally and build quantitative models of their input-output relationships. We then compose these strains into two- and three-strain cascades as well as a four-strain bistable switch and show that experimentally measured consortia dynamics can be predicted from the models of the constituent parts. To further explore the achievable range of behaviors, we perform a fully automated computational search over all two-, three-, and four-strain consortia to identify combinations that realize target behaviors including logic gates, band-pass filters, and time pulses. Strain combinations that are predicted to map onto a target behavior are further computationally optimized and then experimentally tested. Experiments closely track computational predictions. The high reliability of these model descriptions further strengthens the feasibility and highlights the potential for distributed computing in synthetic biology.

**\*For correspondence:**
gseelig@uw.edu (GS);
klavins@uw.edu (EK)

**Competing interest:** The authors declare that no competing interests exist.

## Editor's evaluation

This paper used multiple strains to build gene circuits and demonstrate the modular composition of strain circuits with an automated design strategy to achieve a target behavior from a large space of possible functional circuit architectures. This paper provides synthetic biologists with an alternative solution for the problems of scalability, robustness, and modularity.

## Introduction

The leading paradigm for genetic circuit design is to combine biological parts in a delicate balance within the same cell (*Ellis et al., 2009*; *Kosuri et al., 2013*; *Ottoz et al., 2014*). This approach has resulted in increasingly large genetic circuits that realize functions such as logic gates and circuits (*Moon et al., 2012*; *Bonnet et al., 2013*; *Nielsen et al., 2016*; *Gander et al., 2017*), time pulses (*Gao et al., 2018*; *Guo and Murray, 2019*), incoherent feed-forward loops (*Entus et al., 2007*; *Ellis et al., 2009*), bistable switches (*Chen et al., 2012*; *Huang et al., 2012*; *Yang et al., 2019*; *Barbier et al., 2020*; *Grant et al., 2020*), or oscillators (*Elowitz and Leibler, 2000*; *Tigges et al., 2009*; *Tigges et al., 2010*). Albeit very successful, single cell circuit engineering has limited scalability and robustness because parts cannot be reused, the genetic burden on the cell grows with circuit size (*Nikolados et al., 2019*), and retroactivity (*Del Vecchio et al., 2008*) and component crosstalk

(*Del Vecchio, 2015*) interferes with expected behavior. Furthermore, competition over shared gene expression resources makes unrelated circuits unintentionally coupled (*Zhang et al., 2021*).

An alternative approach is to generate complex behaviors using consortia of cells wherein different cell types perform distinct functions and communicate with each other through chemical signals to realize more complex behaviors. Such division of labor is common in nature and interest has recently emerged in engineering multi-cellular synthetic systems (*Brenner et al., 2008*). Although developed later, synthetic consortia have caught up with single-cell circuits in terms of the complexity of functions that have been realized, generating behaviors such as multicellular time pulses (*Basu et al., 2004*), oscillations (*Chen et al., 2015*; *Danino et al., 2010*), and logic gates (*Regot et al., 2011*; *Tamsir et al., 2011*), bioproduction (*Egbert et al., 2016*), or circuits that use quorum sensing to define social interactions (*Kong et al., 2018*; *Balagaddé et al., 2008*; *Weber et al., 2007*). Furthermore, the emergence of novel orthogonal cross-species signaling molecules (*Billerbeck et al., 2018*; *Du et al., 2020*) and genetic channel-selector devices to multiplex individual signals (*Sexton and Tabor, 2020*) has opened the path for engineering more complex and precise multicellular behaviors.

To date, synthetic multicellular systems were largely designed with specific target behaviors in mind, rather than optimizing modularity of components to make them usable in a large number of contexts. Still, mathematical modeling has demonstrated that multi-cellular computation should easily access a larger space of behaviors: for instance, just three different cell populations can, in theory, generate up to 100 different logic functions (*Regot et al., 2011*) and even bimodality and cell-synchronization (*Thurley et al., 2018*). Furthermore, the intuitive modular nature of cell-to-cell communication should provide a useful tool to rationally design synthetic circuits with predetermined performance. Rational design significantly speeds up circuit assembly (*Chen et al., 2012*; *Chen et al., 2020*) and allows design of global functions from local behaviors (*Salis et al., 2009*; *Carothers et al., 2011*). As a matter of fact, mathematical models have been successful to bridge the gap between individual processes and collective behaviors when applied to ecological interactions between distinct populations of communicating cells (*Shou et al., 2007*; *Momeni et al., 2013*, *Egbert et al., 2016*). However, an example of a model-driven strain selection for multicellular circuit design is currently missing.

Here, we propose a large vocabulary of yeast strains that use chemical signals for cell-to-cell communication and that can be modularly combined to realize a large number of functions (*Figure 1A*). Each strain senses one or two inputs and produces a single output. The transfer function relating inputs and outputs can be either activating or repressing. The output is a fluorescent protein which can be used to read out circuit behavior, another signaling molecule which can be used to connect strains or an enzyme that sequesters or degrades a signal thus disrupting communication. We experimentally characterize each strain, model their dynamics using differential equations and then use these models to predict the behavior of strain combinations (*Figure 1B*). Using a library of 24 strains, we rationally design complex multicellular behaviors, such as bandpass filters, negative and positive feedbacks, time pulses, logic gates, and bistable switches. These behaviors are common in nature and they underlie cell decisions on metabolism (for instance, responding to environmental signals, [*Lee et al., 2002*], utilization of carbon sources, [*Brandman et al., 2005*]) or cell differentiation (*Kueh et al., 2016*; *Duddu et al., 2020*), and have been repeatedly connected to multicellular organization (signal gradients in development) and signaling (auxin pulses in plants). The modularity of composition achieved by our multicellular circuits allow us to design these non-trivial behaviors based on model simulations alone. This qualitative and quantitative precision demonstrated in both time and steady-state experiments further extend circuit design automation (*Nielsen et al., 2016*; *Chen et al., 2020*).

## Results

### Engineering yeast strains for signal sensing, synthesis and depletion

As a first step toward the construction of a vocabulary of modular yeast strains, we selected a set of signals to enable cell-to-cell communication and then optimized chassis strains for signal sensing, synthesis, and depletion. We selected the plant hormone auxin (we will also refer to it as IAA, indole-3-acetic acid) and the yeast hormone $\alpha$-factor as signaling molecules and additionally used the mammalian hormone $\beta$-estradiol as an inducer.

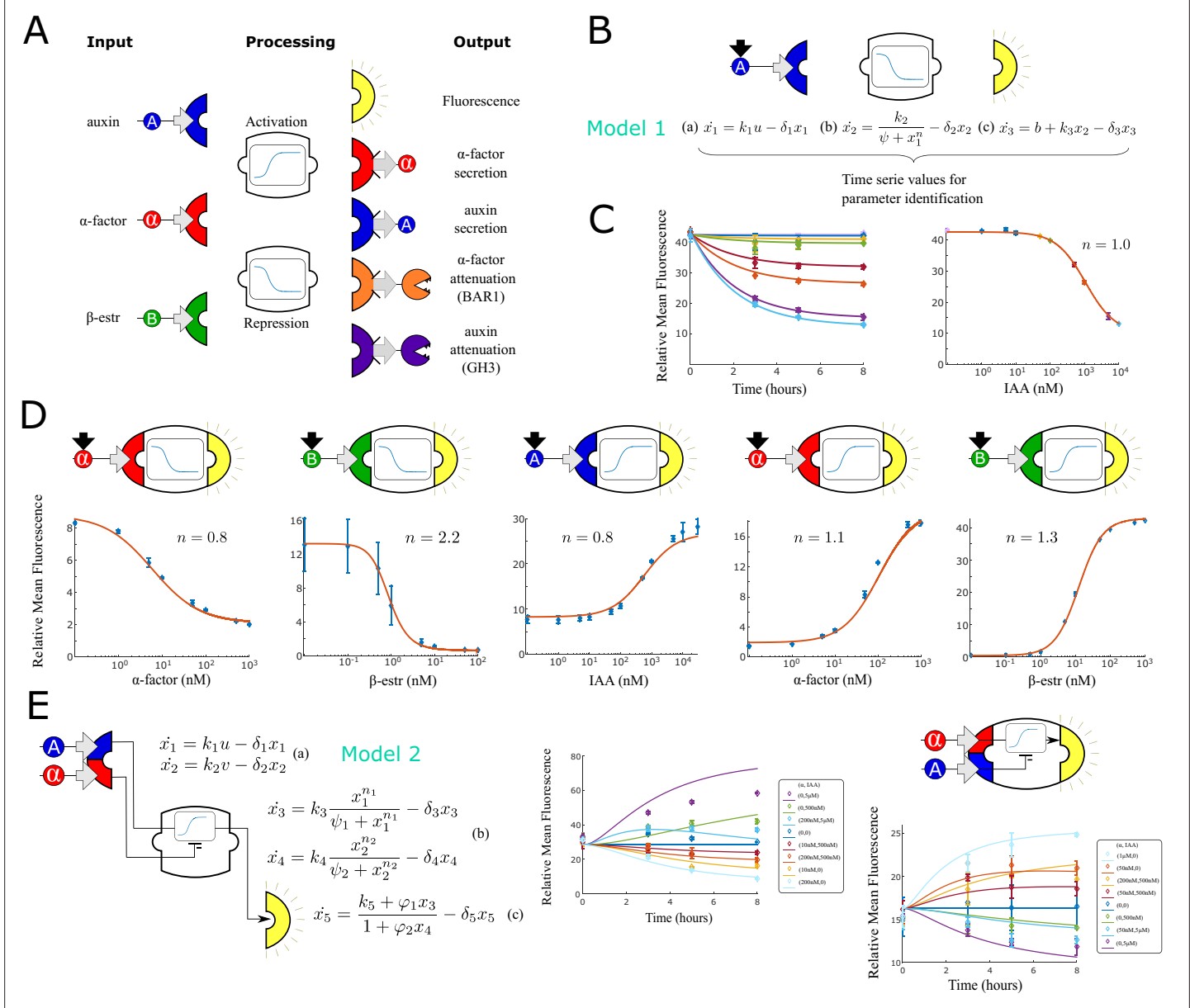

**Figure 1.** Modular components for engineering multi-cellular signaling circuits. (**A**) Three input signals, two transfer functions and five outputs are used to assemble 24 distinct strains. (**B**) Differential equations are used to model and predict behavior of all strains and strain combinations in the paper. In this example, a model (Model 1) and symbolic representation for a fluorescent reporter repressed by high auxin concentration are shown. (**C**) Left: Time course fluorescence data for different auxin concentrations for the sensor strain shown in (**B**). Full lines are model simulations. Right: End point fluorescence data is shown as a function of auxin concentration color-matching the time series on the left. The steady-state simulation is shown in orange. (**D**) Symbolic representation, steady state data and model fit for all other single input reporter strains. (**E**) Modeling framework (Model 2) and fluorescence kinetics data for two different two-input reporter strains. Error bars represents the s.d. of three biological replicates.

The online version of this article includes the following source data and figure supplement(s) for figure 1:

**Source data 1.** The data used for plotting *Figure 1* and supplements.

**Figure supplement 1.** Effect of PGP1 on auxin secretion and sensing.

**Figure supplement 2.** Auxin conjugation to Asp mediated by the GH3 protein in yeast.

**Figure supplement 3.** Mass spectrometry data for IAA-Asp synthesis from IAA in GH3-expressing strains.

**Figure supplement 4.** Circuit pathways for repressing strains using α-factor and β-estradiole as inputs.

**Figure supplement 5.** Pathway optimization for IAA-sensing strains.

**Figure supplement 6.** Strains constructed but not used in this study.

*Figure 1 continued on next page*

*Figure 1 continued*

**Figure supplement 7.** Orthogonality between signaling molecules.

The $\alpha$-factor pathway has been subject of extensive studies and mechanisms for sensing through the surface receptor STE2, synthesis by the MF$\alpha$1 gene and degradation through the BAR1 protease are well understood and have been engineered to generate a wide range of behaviors (*Youk and Lim, 2014*; *Groves et al., 2016*; *Shaw et al., 2019*). To have a baseline strain that does not interfere with signaling, we knocked out the native BAR1 gene as in *Youk and Lim, 2014* to prevent $\alpha$-factor degradation. Moreover, to account for growth-arrest induced by $\alpha$-factor, which affects gene expression on a large scale, we knocked out FAR1 a protein that contributes to arresting the cell cycle at G1Chang and *Chang and Herskowitz, 1990*, and constitutively expressed POG1, a protein that promotes growth-arrest recovery (*Leza and Elion, 1999*). Unexpectedly, we detected no $\alpha$-factor output from strains that both sense and secrete $\alpha$-factor. We suspected that this was caused by the surface receptor STE2 internally binding to $\alpha$-factor. It has been shown that, upon binding to $\alpha$-factor, STE2 undergoes endocytosis and then shares the secretory pathway of $\alpha$-factor itself (*Schandel and Jenness, 1994*). We solved this problem by overexpressing STE2 on a pGPD promoter *Sun et al., 2012* in these strains, aiming to have some protein copies escaping this interaction.

Next, we turned to the optimization of components for Auxin sensing, synthesis, and elimination. In prior work from our group, we (*Khakhar et al., 2016*; *Pierre-Jerome et al., 2014*) developed an auxin-responsive transcription factor using a chimeric dCas9-Aux/IAA protein regulation. In the presence of Auxin, the Aux/IAA degron part of the protein binds to the auxin signaling F-box protein (a modified TIR1 for this study) and acts as part of an E3 ubiquitin ligase to catalyze ubiquitination and degradation of the Aux/IAA-degron-containing protein. Similarly, we constructed a synthetic auxin synthesis pathway in yeast (*Khakhar et al., 2016*). We demonstrated conversion of the precursor molecule IAM (indole-3-acetamide) into IAA through expression of the IaaH gene in yeast. Here, we amplified Auxin secretion and thus effectively the strength of the signal produced, through integration of the auxin-efflux pump ABCB1/PGP1 from *A. thaliana* into our yeast strains, as previously reported in *Geisler et al., 2005*. We measured a significant increase in the auxin-synthesis yield as measured by a neighboring IAA-detecting cell when PGP1 was integrated (*Figure 1—figure supplement 1*).

Unlike for $\alpha$-factor, a depletion mechanism for Auxin had not yet been reported in yeast. To create an IAA depletion mechanism, we thus selected the plant protein GH3.3 that has been shown to conjugate IAA to aspartic acid, forming the signaling-inactive IAA-Asp. GH3.3 is part of a family of proteins that encode IAA-amino synthetases, which have been reported to control auxin homeostasis (*Staswick et al., 2005*). To test whether GH3.3 or related proteins could be used to inactivate Auxin in yeast, we first expressed codon-optimized versions of GH3.3 and GH3.6 from *A. thaliana* and *C. papaya* from a highly expressed pGPD promoter. We then tested the IAA to IAA-Asp conversion rate using mass spectrometry and found that GH3.3 from *A. thaliana* had the higher efficiency (*Figure 1—figure supplement 1*). Finally, we tested that IAA-Asp does not activate the IAA-mediated degradation pathway in yeast, by adding 10 µM of IAA-Asp in an auxin-sensor yeast culture and detecting no variation in fluorescence (*Figure 1—figure supplement 1*).

## Establishing a vocabulary of parts for cell to cell communication

Having established conditions for efficient signal sensing, synthesis and inhibition, we combined signals with activating and repressive transfer functions to create a vocabulary of strains. Transfer functions that use $\alpha$-factor as input are mediated by the transcription factor (STE2) that either directly induces expression of the gene of interest (activation) or induces expression of a repressor that inhibits the output (repression). Similarly, $\beta$-estradiol binds and activates a transcription factor (ZEV4, *McIsaac et al., 2013*) that either directly activates gene expression (activation) or induces expression of a repressor and inhibits output synthesis (repression). For both signaling molecules, we chose dCas9 fused with the repressor domain Mxi1 (*Gander et al., 2017*) as repressor (*Figure 1—figure supplement 1* for the full pathways).

To induce activation or repression with auxin, we build on the same auxin-mediated degradation pathway used for auxin sensing above. Specifically, activation results from degradation of a repressor, while repression results from degradation of an activator. We chose a dCas9-Mxi1-auxin degron

(*Khakhar et al., 2016*) fusion as a repressor and used a dCas9-VP64-auxin degron fusion as the activator (*Figure 1—figure supplement 1*).

We conducted extensive pathway optimization to increase the separation between high and low expression levels and IAA sensitivity, using a mechanistic model to explore the parameter space and guide the genetic engineering (*Figure 1—figure supplement 1*). We adopted the model proposed in *Pierre-Jerome et al., 2014*, where each parameter easily translates to a biological function, and performed parameter sensitivity analysis with respect to fold change between the baseline and the Aux/IAA-induced fully repressed state. We then selected the top six highest scoring parameter perturbations, designed circuit variants that reflected those changes, and tested them resulting in a good agreement with our predictions (*Figure 1—figure supplement 1*). Guided by the model, we combined five of the tested circuit variants to increase the fold change from 1.3-fold (sensor from *Khakhar et al., 2016*) to 3.1-fold: we used this final circuit for all the repressive strains, swapping the fluorescent reporter gene with the output gene for non-sensor strains. For the auxin activating strains, we used a similar model-driven approach to rationally design the activating pathway novel to this paper, obtaining a threefold-change activation.

Combining the three different input signals, the activation/repression circuits and output secretion, we built and tested all the possible combinatorial designs presented in *Figure 1A*. The strains that sense $\beta$-estradiol and repress expression of BAR1 and GH3.3, the strain that senses $\alpha$-factor and represses expression of GH3.3, and the strain that senses IAA and represses BAR1 expression are shown in *Figure 1—figure supplement 1* and not used below, since their response was too slow to produce meaningful results. Of the remaining 24 strains, 6 sensor strains express GFP in response to the three input signals (three activators and three repressors), 12 strains (six activators and six repressors) synthesize a signaling molecule, and 6 strains act as signal attenuators (expressing BAR1 or GH3.3). Four of these 24 strains sense and secrete the same signaling molecule ($\alpha$-factor or IAA, 'positive' or 'negative' feedback strains). Finally, two strains express repressors of their own input ($\alpha$-factor expressing BAR1 and IAA expressing GH3), also describing a negative feedback topology (*Figure 1—figure supplement 1*).

## Sensor strain characterization

For sensor strain characterization, we collected time series data for eight different input concentrations. Input concentrations were selected to fully cover the sensor dynamic range for model fitting. We normalized fluorescence data by cell size and took the mean of the histogram as in *Groves et al., 2016* and then subtracted background fluorescence. Each measurement in the figure is an average of three experimental repeats (error bars representing the standard deviation).

With a scalable and modular system in mind, we fitted a set of three ordinary differential equations (ODEs) with eight parameters for each strain to describe input sensing ($x_1$), signal processing ($x_2$) and fluorescence output synthesis ($x_3$) (*Figure 1B*, Model 1). Signal processing (activation or repression) is modeled with a simple Hill function (Model 1b), which naturally incorporates signal saturation. Input sensing and output synthesis are modeled as linear ODEs (Model 1 a and c). A constant term in the last ODE accounts for background promoter activity. For instance, for the auxin sensor in *Figure 1B*, x represents the TIR1-auxin complex concentration, $x_2$, the dCas9-VP64-auxin degron population, and $x_3$, GFP concentration. These simple models capture the system dynamics, with the benefit of being easy to fit and analytically approachable. Parameters were fitted independently for each experimental repeat to obtain a mean and a standard deviation for the Hill coefficient. We also separately fitted a Hill curve to an average of the three experimental repeats and the resulting Hill coefficient ($n$) is reported in the figures and used for simulations (*Figure 1B*). The sensor strains range in sensitivity depending on the input. The $\beta$-estradiol sensors respond to inputs concentration ranging from 0.1 to 100 nM with an $EC50$ of 0.5 ± 0.0 nM for repression and 12.6 ± 0.9 nM for activation. $\alpha$-factor sensors are sensitive to input concentrations ranging from 1 to 500 nM range with an $EC50 \sim 6.0 \pm 0.4$ nM for activation and 89.0 ± 6.6 nM for repression. Finally the IAA sensor is least sensitive and responds to inputs ranging from 5 nM-10 uM withe an $EC50 \sim 964.8 \pm 93.8$ nM for activation and 276.5 ± 8.8 nM for repression. The Hill coefficients for these sensors vary between 0.8 (3-repeat interval: 0.8 ± 0.0, $\alpha$-factor repression) and 1.0 (1.2 ± 0.02, $\alpha$-factor activation), 2.2 (2.1 ± 0.3, $\beta$-estradiol repression) and 1.3 (1.25 ± 0.0, $\beta$-estradiol activation), 1.0 (1.0 ± 0.4, auxin repression) and 0.8 ($0.8 \pm 0.0$, auxin activation) consistent with previously reported values for $\beta$-estradiol repression (dCas9-Mxi1 confers

stronger and more consistent repression than dCas9 alone, leading to ultrasensitivity, *Gander et al., 2017*) and $\alpha$-factor activation (*Groves et al., 2016*). Finally, the sensors achieve an ON/OFF fold-change that ranges between a minimum of 3 (IAA activating GFP) and a maximum of 42 ($\beta$-estradiol activating GFP). We tested signal orthogonality by adding pairwise combinations of inducers at saturation concentration to single-input repressive sensors (*Figure 1—figure supplement 1*). Variations across the treatments are contained within 11% of the nominal value for signal concentrations within the range used for circuit experiments.

In addition to single-input sensors, we constructed two sensor strains that respond to both $\alpha$-factor and IAA. In the first strain, $\alpha$-factor induces fluorescence expression and IAA represses it, while in the second strain, IAA induces fluorescence expression and $\alpha$-factor represses it (*Figure 1E*).

We collected data points for eight different input combinations taken at five time points up to saturation. The output signal is monotonic with respect to each input and the input functional range is similar to the one measured for the correspondent strains that respond to only one of the two inputs.

To model these strains, we used a simple extension of our previously introduced models with five ODEs (*Figure 1E*, Model 2): two ODEs model input sensing (a), two model input processing (b), and one ODE (c) combines the signals according to their activating or repressing nature and defines output synthesis. These models fit the experimental data even when the output is non-monotonic over time.

## Assembling modular, tunable and easily-extendable circuits using cell-to-cell communication

To determine the potential to build biological circuits using cell-to-cell communication, we experimentally tested if communication occurs between strains that secrete an output and their corresponding sensor strains. For example, a strain that produces auxin in response to $\alpha$-factor sensing was grown in co-culture with a sensor that switches off GFP expression in response to auxin (*Figure 1B*). The two strain populations were added at the same concentration and the fluorescent output was measured at steady state (10 hours after induction). The experimental data is consistent with a negative $\alpha$-factor sensor as expected (the more $\alpha$-factor, the lower the fluorescent signal, as seen in *Figure 2A*).

Since the core genetic circuit of this sender strain is identical to the $\alpha$-factor repressing sensor, we tested if the mathematical model previously fit to the sensor data preserves its predictive power. We re-fit only the 3 output parameters to account for the fact that the output is now auxin rather than GFP. The output of the sender cell model was used directly as input of the sensor cell model. To test the hypothesis that strains that have common input/processing parts share parameters and model structure, we collected two datasets on two different experiments composed of four data points each: we used one for fitting (yellow dots, first experiment), and the other for validation (orange dots, second experiment, *Figure 2A*). A time-series at EC50 input concentration was also used for fitting (not present in the figure).

Intuitively, we expected that higher initial sender cell concentration would result in an overall higher concentration of their output signal over the same time scale. Most importantly, we wanted to know if this output 'gain' can be predicted by our models so that we could use it to tune circuit behavior. We modeled this effect with a factor $K$ multiplied by the output signal, where $K$ is the fold-change with respect to the standard initial concentration: we represented this gain using the same iconography used in electronic circuits (*Figure 2B*). Our predictions matched closely with data collected using 5 X and 10 X the original concentration of sender cells (green and blue lines) using the same strain co-culture as in the previous panel (orange line). Notice that the fold-change, as we just defined it above, is not equivalent to the ratio between the strains. Hence, difference in growth rates are not relevant for model predictions provided that there is no dilution or competition for resources, which is minimal in these experiments (the strains are kept in log-phase throughout the experiments and their concentrations are kept below saturation).

We further tested if we could modulate the concentration of signaling molecules by removing it from the system through BAR1 and GH3.3-expressing strains (*Figure 2C*). We modeled signal degradation as a first-order Hill repressing function, where the output of the sender cell acts as a negative regulator. As before, only the three output parameters of the sender strain were fitted using receiver cell fluorescence data. Finally, we tested all the sender-receiver pairs in our vocabulary with the exception of those generating positive or negative feedback. All activating strains function within the sensor range of their receiver strains (*Figure 2D*), and the models correctly fit or predict the data. On the

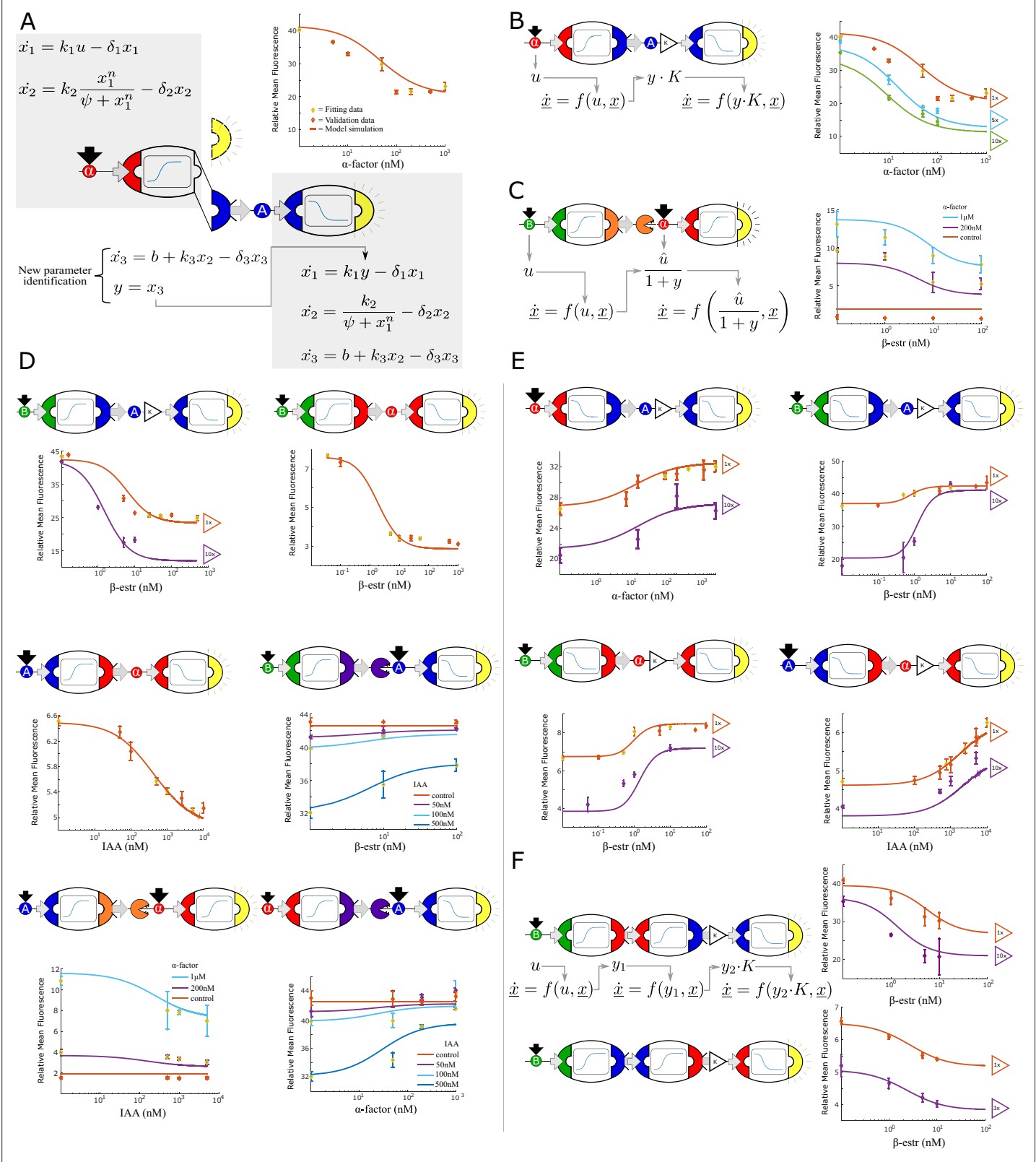

**Figure 2.** Multi-strain signaling cascades. (**A**) Symbolic representation of a two-strain cascade that uses α-factor as the input and auxin as an intermediate signal. To model multi-strain cascades, models for individual strains are concatenated and only the last differential equation of the first model is fit: the GFP output (dashed yellow semi-circle) is substituted with IAA (solid blue semi-circle). End-point fluorescence data and fit are shown for the example cascade. (**B**) By varying the concentration of the upstream strain, the strength of the signal seen by the downstream strain can be

*Figure 2 continued on next page*

*Figure 2 continued*

predictably controlled. This change in signal is modeled with a single parameter K. Right: data and model predictions for three experiments with the same strains but varying concentrations of the upstream strain. (**C**) Symbolic representation, model and data for a two strain cascade wherein the upstream strain removes a signal rather than secreting it. (**D**) Symbolic representation, model and data for all two strain cascades where the upstream input is activating the production of the intermediate signal. (**E**) Symbolic representation, model and data for all two strain cascades where the upstream input is repressing the production of the intermediate signal. (**F**) Symbolic representation, model and data for three-layer signaling cascades. Error bars represents the s.d. of three biological replicates.

The online version of this article includes the following source data for figure 2:

**Source data 1.** The data used for plotting *Figure 2*.

other hand, the output of repressing strains (*Figure 2E*) did not fully cover the input range of their receiver strains, as seen in the limited fold-change of the fluorescent signal. Models suggested that increasing the sender strain population to 10-fold its original value would produce a more noticeable response, which we successfully verified experimentally (purple dots and lines, *Figure 2D and E*).

After the two-strain combinations, we also verified the predictive power of our model on two 3-strain chains with different topologies and different strain stoichiometries (*Figure 2F*). Here too, the simple model strategy we outlined earlier captured the overall dynamics even when different strain concentrations were used. These results support the hypothesis that multicellular circuits behave like a sum of their individual parts (modularity), are easy to modulate (whether through altering initial strain concentrations or signal degradation) and can be extended to longer chains.

## Increasing nonlinear response using external positive feedback

Thus far, we explored ways to simulate and design multicellular circuits with tunable gains to obtain monotonic, quasi-linear dynamic systems with a single equilibrium point. To extend the range of observable behaviors and generate non-linear responses to the inputs, we used positive feedback strains that sense and secrete the same signaling molecule (*Figure 3A*).

To highlight the increase of nonlinear response, on top of fitting models to the positive feedback strains (as in *Figure 2A*), we also re-fitted the sensor strains to estimate a new Hill coefficient (much as in *Shaw et al., 2019*). We then plotted these fitted curves, the data points and the two Hill coefficients for both the feedback system and the nominal response (in this case, the sensor alone). In both cases, the positive feedback increased the nonlinearity of the response (from $0.8 \pm 0.0$ to $1.5 \pm 0.1$ and from $1.0 \pm 0.4$ to $1.2 \pm 0.3$ for the $\alpha$-factor and the auxin case respectively). We also considered positive feedback circuits that operate through double repression (*Figure 3B*). In this case, we tested topologies where either $\alpha$-factor represses BAR1 synthesis, or auxin represses GH3.3 synthesis. Ideally, at low signaling molecule concentration, signal degradation through BAR1 or GH3.3 is predominant so there is no fluorescence response in the receiver cells. But at high input concentration, the degradation pathway is switched off and the input is free to reach the receiver cells. As in *Figure 3A*, we measured the Hill coefficients of these circuits and reported an increase in nonlinear response (from $0.8 \pm 0.0$ to $1.0 \pm 0.0$ and from $1.0 \pm 0.4$ to $1.4 \pm 0.6$ for $\alpha$-factor and auxin circuits respectively).

## Constructing a multicellular bistable switch

We next turned to the construction of a bistable switch circuit. We opted to use a mutual-repression topology to generate bistability (*Gardner et al., 2000*; *Oyarzún and Chaves, 2015*). For a first design, we combined the strain that senses $\alpha$-factor and represses auxin synthesis with the strain that senses auxin and represses $\alpha$-factor synthesis. Here, the main state variables are signaling molecule concentration in the media rather than the internal state of the cells (specifically, high auxin/low $\alpha$-factor and low auxin/high $\alpha$-factor). However, both model simulation and experimental data showed that this circuit can generate only a single equilibrium, independently of the strain stoichiometry (*Figure 3—figure supplement 3*). This result is not unexpected given the low Hill coefficients of the two repressive circuits (0.8 and 1, respectively). Rather than re-designing the gene regulatory circuits internal to these strains to be more suitable for a bistable switch, we decided to take advantage of the composable nature of the system and incorporate more strains to the architecture.

As shown in *Figure 3B*, combining strains that work cooperatively increases the Hill coefficients. Hence, to boost non-linearity, we added two strains to the mix that induce signal degradation: a strain that senses $\alpha$-factor and synthesizes GH3 and one that senses auxin and synthesizes BAR1 (*Figure 3C*).

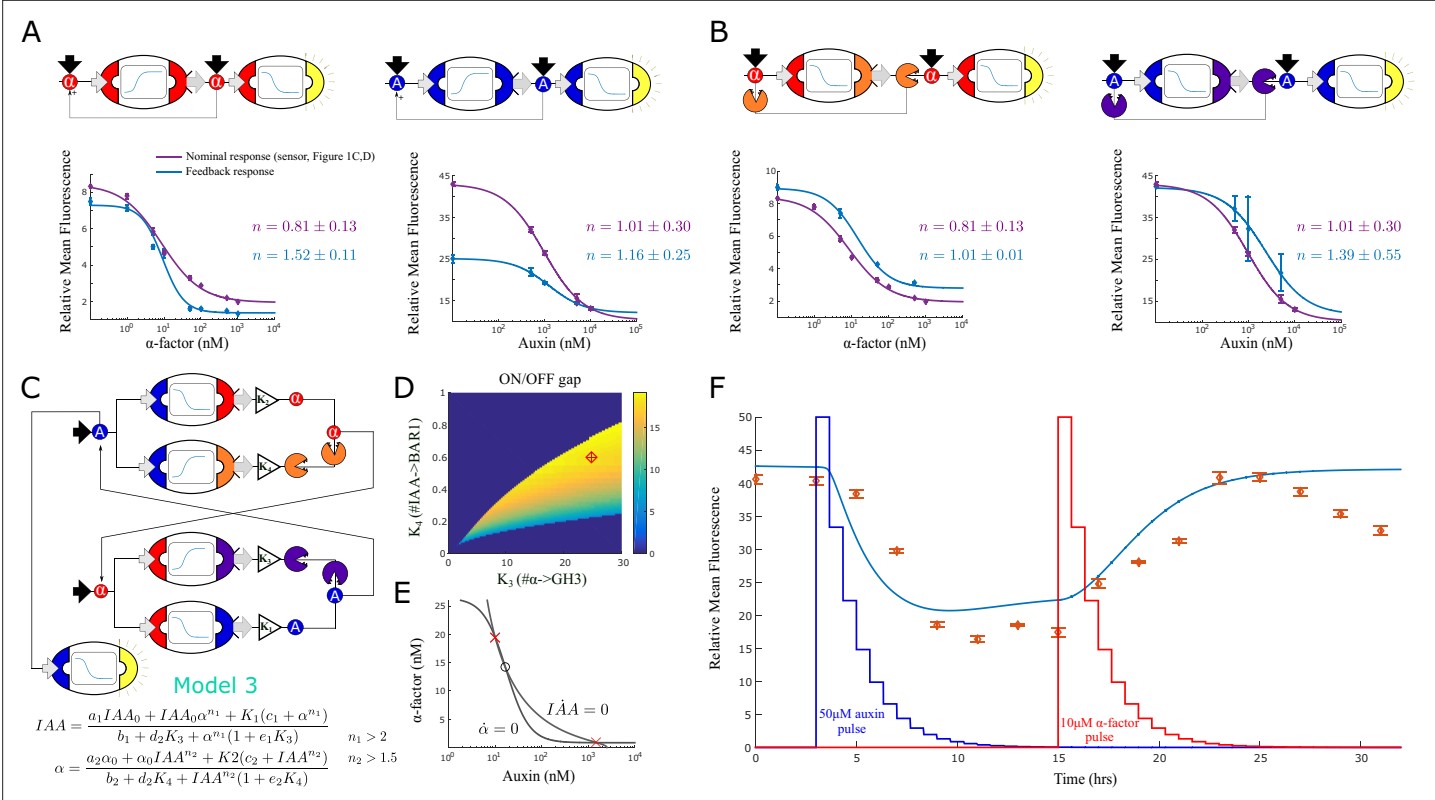

**Figure 3.** Feedback strains and bistability. (**A**) Symbolic representation, model and data for two self-activating positive feedback circuits using alpha factor (left) and auxin (right) as the activating signal. (**B**) Symbolic representation, model and data for two positive feedback circuits that act through double repression. (**C**) Architecture and model for a bistable switch assembled from four separate strains. (**D**) Area of bistability as a function of strain concentrations, where the dark blue color represents one stable equilibrium and color shades report predicted fold-change between ON and OFF states. The coordinates of the red diamond are the concentrations chosen for experimental testing. (**E**) For the chosen solution, the nullclines show the IAA and alpha-factor concentration for the three equilibria ($\times$ = stable, $o$ = unstable) (**F**) Experimental data for the bistable switch. Full line is Model 3 simulation, and exogenous auxin and alpha-factor concentrations are reported as a function of dilution over time, depicted in the figure as jagged blue and red lines (the peaks are normalized to simplify their graphical representation). Error bars represents the s.d. of three biological replicates.

The online version of this article includes the following source data and figure supplement(s) for figure 3:

**Source data 1.** The data used for plotting *Figure 3* and supplements.

**Figure supplement 1.** Mutually repressing strains do not generate bistability.

**Figure supplement 2.** Model explanation for the stable-to-unstable shift.

**Figure supplement 3.** Growth rate of the five strains in the bistable switch.

The resulting system can still be seen as two modules that repress each other's activity: $\alpha$-factor lowers auxin concentration (through pathway repression and GH3 expression) and auxin reduces $\alpha$-factor concentration (through pathway repression and BAR1 expression). We studied this new system with a steady-state model for auxin and $\alpha$-factor concentrations (*Figure 3C*, Model 3 and Appendix 1 for model derivation). The model suggests that one of the two Hill coefficients ($n_1$) is higher than 2, a necessary condition for the existence of more than one equilibrium.

We investigated the existence and properties of the equilibria while varying the individual concentrations of the four strains in the circuit: these variations are captured by the four $K$ parameters in the model, representing the gains of the four strains in the circuit as explained earlier. Using the model, we identified a range of concentrations predicted to result in multiple equilibria (for the analysis, see Appendix 1). For further investigation, we picked a set of concentrations that maximize the distance between equilibria such that the equilibria are robust to small variations in strain concentrations (red diamond, *Figure 3D*). This solution generates two nullclines that intersect three times, resulting in 2 stable (red crosses) and 1 unstable (black empty circle) equilibrium as expected (*Figure 3E*).

We tested this model-guided design experimentally (*Figure 3F*). Strains were mixed according to the chosen concentrations and an auxin negative sensor strain was added. Upon reaching log phase (Time 0 in *Figure 3F*), samples were diluted at regular time intervals to prevent the cells from saturating (see the Materials and methods section). To alternate between the states, we exogenously added first auxin (at 3 hr) and then $\alpha$-factor (at 15 hr) and let dilution reduce their concentration to below detectable levels for our sensor (at 12 and 24 hr, respectively).

Model 3 simulation and data largely agree, although there is a lag between the two. We hypothesize that the lag is due to a signaling delay between the strains that was not fully captured by the models in this five-strain circuit (four strains for bistability plus one auxin sensor). More importantly, the 'high' equilibrium is stable in the first 3 hr of the experiment, but seems to become unstable after 25 hr, as shown by a negative trend in the data toward the 'low' equilibrium. Modeling suggests that a small increase in the $K_4$ gain (strain: IAA expressing BAR1) would lead the system to a single 'low' equilibrium (*Figure 3—figure supplement 3*). This could occur if the corresponding strain grows slightly faster than the other strains (although, in practice, we could not detect any growth difference between the single-input/single-output strains using an ANOVA test, *Figure 3—figure supplement 3*). An alternative explanation is that metabolic load affects the growth rates depending on strain circuit activity (*Sadeghpour et al., 2017*; *Zhang et al., 2020*). Accordingly, active strains grow more slowly than inactive ones, affecting their intended concentrations over time, which could result in leaving the bistability region. To test this hypothesis, we introduced metabolic load to our models (Appendix 1). Following *Sadeghpour et al., 2017*, we assumed metabolic load to be linearly proportional to (normalized) gene expression This dependency is captured by a parameter $0 \leq \rho \leq 1$, where $\rho = 0$ represents no impact and $\rho = 1$ represents a high impact on growth rate (no growth at maximum gene expression). Simulations show that a behavior qualitatively similar to the loss of stability after 25 hours is possible for high metabolic load ($\rho \geq 0.5$), although at a later time than the experimental data (~36 hrs, *Figure 3—figure supplement 3*) according to this model representation. Moreover, since growth rates are statistically identical for active and inactive strains (*Figure 3—figure supplement 3*), we concluded that metabolic load is unlikely to significantly affect our bistable circuit. This result is consistent with previous work supporting robustness of the repressive circuit topology to growth feedback (*Zhang et al., 2020*).

## Automated design of strain circuits to generate logic gates

In order to expand our target behaviors, we developed an automated approach to select circuits using the twenty-four strains introduced above. Specifically, we simulated all possible strain combinations for networks of size 2, 3, and 4 strains using Models 1 and 2. For each network, we simulated the system response over 12 hours to all possible combinations of $\alpha$-factor (in the discretized range of $[0 > 1 > 5 > 10 > 50 > 200 > 1000]$ nM), auxin ($[0 > 100 > 500 > 1000 > 5000]$ nM), and $\beta$-estr ($[0 > 1 > 5 > 10 > 100]$ nM). The ranges were chosen to properly sample the operational range of the strains (see *Figure 1C, D and E*).

Next, we screened our simulation space for steady-state behaviors whose profiles resemble AND, OR, NAND, or NOR logic gates. For each strain combination, we restricted the steady-state behavior space to simulations obtained using combinations of the highest input concentrations ($\alpha$-factor = 1000 uM, auxin = 5000 μM, $\beta$-estr = 100 nM) or no input to match the $[01]$ logic table input entries. This selection resulted in 120 combinations from the 2-node networks, 560 from the 3-node networks, and 1,820 from the 4-node networks. After normalization to allow for comparison between different sensors (Appendix 2), each of these combinations was scored according to how well they match one of the logic gate truth tables. If the predicted output for one of the expected OFF states was higher than the output for one of the expected ON states then the metric would return a value below 1 labeling that strain combination to be a poor logic gate realization. On the other hand, values above 1.0 imply a match between the output vector and the target truth table: the higher the value, the better the separation between the ON and OFF state.

We first applied this method to identify circuit topologies that generate AND gate profiles (*Figure 4A*). Each network topology is represented as a diamond, color-coded according to the network size (blue for 2-node, red for 3-node and yellow for 4-node networks), and divided in 4 groups according to the sensor strain reporter. Each network was scored, and its value reported on the x-axis.

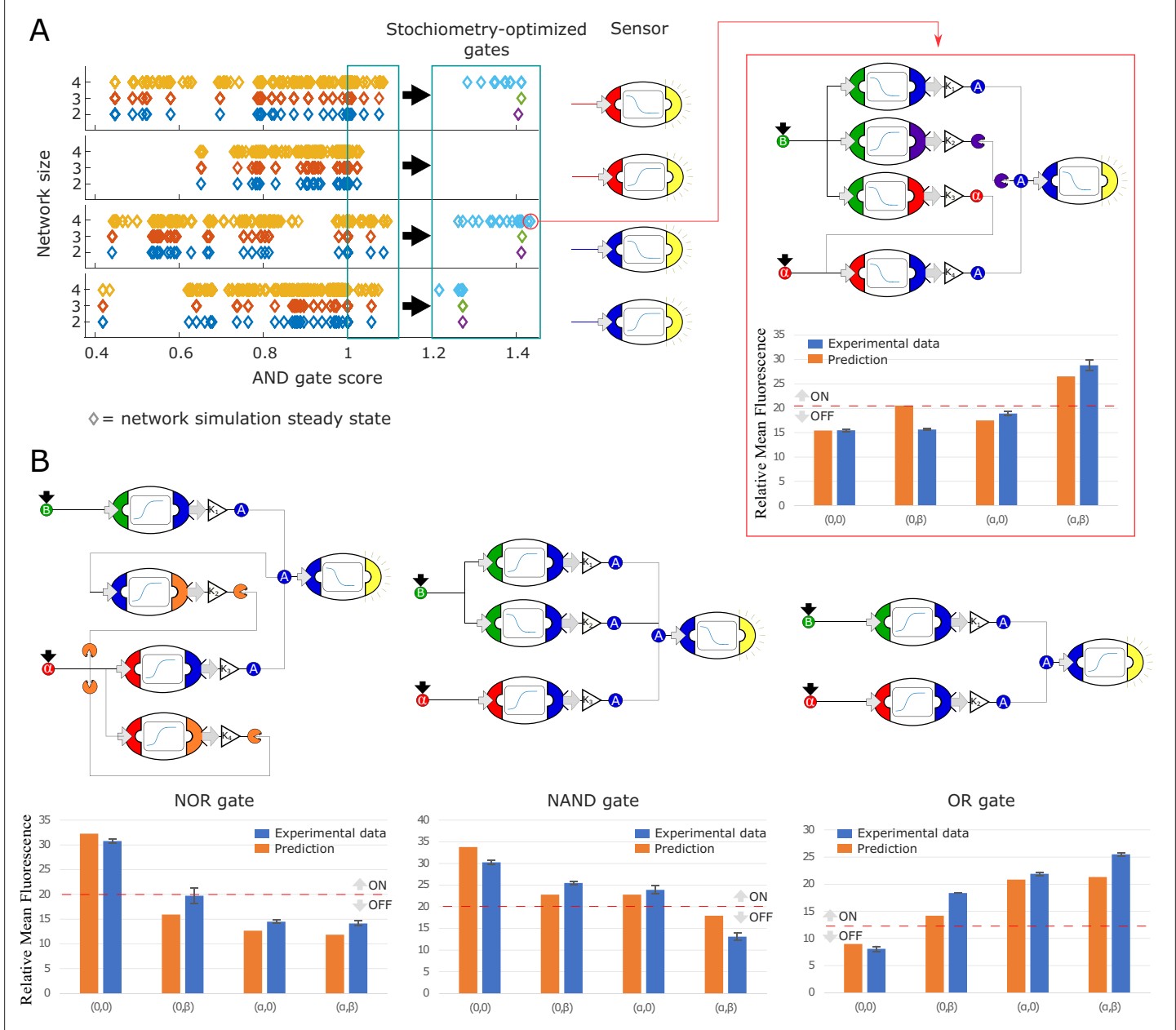

**Figure 4.** Model-generated implementation of Boolean logic gates. (**A**) We used an automated search algorithm to screen all possible strain combinations up to size four (and including exactly one sensor strain) for their ability to realize a set of logic functions. In the example, each strain combination is scored according to how close the combination is to realizing an ideal Boolean AND gate. Each colored diamond corresponds to one strain combination. Circuits consisting of two strains (+ a sensor) are shown in blue, three-strain gates are shown in red, and four-strain gates in yellow. All strains in these simulations are at equal stoichiometry. Higher scores indicate more AND-like behavior. The top-performing strains (score >1, teal box) are selected for further computational optimization of strain stoichiometry. Optimized strain combinations have higher AND scores. Optimized circuits consisting of two strains are shown in light blue, three-strain gates are shown in green, and four-strain gates in purple. Simulations are separated according to which sensor strain is used. A specific high-scoring four-strain combination was chosen for experimental testing (red box). The experimental data (blue bars) show good quantitative agreement with the predictions (orange bars). (**B**) Similar optimization procedures to the ones shown in (**A**) were used to identify strain combinations that realize NOR, NAND, and OR logic functions. Example implementations, model predictions and experimental data are shown for all three logic functions. Error bars represents the s.d. of three biological replicates.

The online version of this article includes the following source data and figure supplement(s) for figure 4:

**Source data 1.** The data used for plotting *Figure 4* and supplements.

**Figure supplement 1.** Optimal circuits, simulation, and experimental realization of the AND gate for 2, 3, and 4-node topologies.

*Figure 4 continued on next page*

*Figure 4 continued*

**Figure supplement 2.** Optimal circuits, simulation, and experimental realization of the OR gate for 2, 3, and 4-node topologies.

**Figure supplement 3.** Optimal circuits, simulation, and experimental realization of the NAND gate for 2, 3, and 4-node topologies.

**Figure supplement 4.** Optimal circuits, simulation, and experimental realization of the NOR gate for 2, 3, and 4-node topologies.

Next, we selected all the network topologies that scored higher than 1.0 and performed an optimization step. Optimization aimed to maximize the target metric using strain concentration (gains) as optimization parameters (*Figure 4A*, left panel). Finally, we picked the highest scoring topology for experimental testing. The best AND gate was a 4-node network topology that used $\alpha$-factor and $\beta$-estradiol as inputs and a negative auxin sensor to determine the output.

This strain combination included two strains repressing auxin output in the presence of $\alpha$-factor and $\beta$-estradiol respectively. These strains by themselves should ideally generate an AND gate in concert with the negative auxin sensor. The other two strains improve performance of the AND gate, likely by reducing the effect of leaky auxin synthesis from the $\alpha$-factor-sensing/auxin-repressing strain. The experimental realization of this circuit shows separation between the ON and OFF states. In fact, the data (blue bars) seem to slightly outperform the predictions (*Figure 4A*, right panel).

We repeated this same procedure for NOR, NAND, and OR gates (*Figure 4B*). The optimal NOR gate is also a four-node network, while the optimal NAND gate is a three-node network, and both use the same auxin sensor as the optimal AND gate. The bar separating ON and OFF states as defined for the AND gate holds for all the gates sharing the same sensor. The optimal OR gate is the naive realization with two strains synthesizing auxin from different inputs and an activating an auxin sensor. All the gate profiles are close to their predicted values from simulation, showing the high degree of modularity of our vocabulary of strains, even for complex systems with internal feedback as the NOR gate architecture. For each gate, we also tested the optimal realization of the strain combinations for the two remaining network sizes with similarly positive results (*Figure 4—figure supplement 4*).

## Identification of circuit designs for time pulses and band pass filters

We next extended our automated design strategy to circuits that either generate time pulses or that acts as band pass filters on the input signal concentrations. Starting from the simulation dataset of all possible strain combinations, we selected all those that displayed non-monotonic behaviors (having at least one local maximum/minimum) as a function of time or as a function of the steady state input concentration. We defined a non-monotonicity metric as the distance between the local maximum/minimum and the maximum/minimum between the initial value and the final value of the series. Higher values of the metric hint to more pronounced non-monotonic behaviors, while 0 implies that no local minimum/maximum is present (*Figure 5A*). We then selected the top six candidates (one for each sensor type in *Figure 1B, D and E*, excluding the $\beta$-estradiol ones), and performed optimization to maximize the metric using cell concentrations and input concentration as parameters. Finally, we experimentally tested the top two topologies overall for both time pulses and steady-state band-pass filters. It is worth noticing that only 0.52% of all possible topologies across network sizes generated non-monotonic behaviors in time (of about $10^6$ in total, *Figure 5A*) and only 0.32% at steady state (of about $10^5$ in total, *Figure 5B*). Hence, a 'brute force' experimental approach to test all possible strain combinations would be evidently out of reach.

The highest performing time pulse topology (*Figure 5C*) is induced by $\alpha$-factor that activates both fluorescence expression and auxin synthesis. In turn, auxin induces BAR1 production, which degrades the exogenous $\alpha$-factor signal: unsurprisingly, this is an incoherent feed-forward loop, type 1 (as in *Mangan and Alon, 2003*). The optimal 'reversed' time pulse, that is, a dip in the output at intermediate times (*Figure 5D*), responds to $\alpha$-factor induction and implements a modified incoherent feed-forward loop type 3, where $\alpha$-factor both represses and activates fluorescences. Model predictions suggested that three different $\alpha$-factor concentrations would generate this behavior at different capacities (0.1 nM, 1 nM and 5 nM $\alpha$-factor).

According to the models, both these nonmonotonic behaviors are a consequence of delay between an activator and an inhibitor pathway.

The low-pass concentration filter (*Figure 5E*) uses very similar components but the sensor with opposite topology. In this case, $\alpha$-factor activates fluorescence expression but also auxin synthesis,

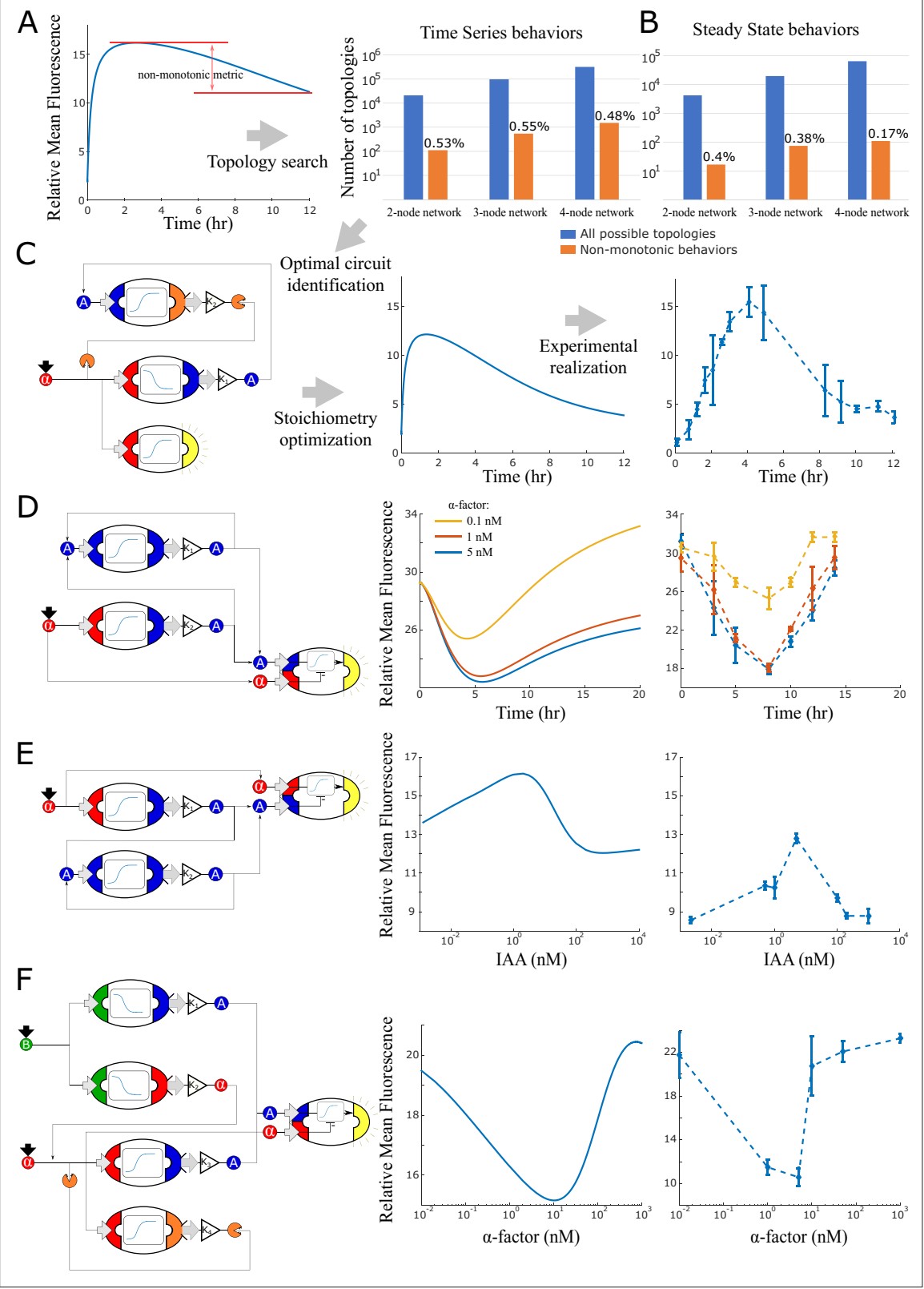

**Figure 5.** Model generated implementations of analog functions. (**A**) An automated search algorithm was used to screen all possible strain combinations up to size four (plus one sensor strain) for non-monotonic behaviors. Specifically, we set up the search to find combinations that result in a pulse as a function of time or as a function of concentration (**B**). Top row: We define a non-monotonicity metric and rank all combinations according to that score. Bar graphs show the total number of possible strain combinations (blue) and the percentage that show non-linear behaviors. Results are

*Figure 5 continued on next page*

*Figure 5 continued*

organized by size of the strain combination and the type of target behavior. (**C**) Strain stoichiometries are optimized for the most promising strains to obtain more extreme maxima or minima. Left: a diagram of the highest scoring circuit for time pulses selected for testing. Center: behavior prediction after circuit stoichiometry optimization. Right: Experimental data for the time pulse circuits is shown. (**D**) Strain combination, model prediction, and experimental data for a system that results in a dip in fluorescence as a function of time rather than a peak. (**E**) Strain combination, model prediction and experimental data for a system that generates a peak at intermediate auxin concentration thus realizing a band-pass filter. (**F**) Strain combination, model prediction and experimental data for a system that generates a dip at intermediate alpha-factor concentrations. This system realizes a 'band-stop' filter, which is a combination of a low and a high-pass filter. Error bars represents the s.d. of three biological replicates.

The online version of this article includes the following source data for figure 5:

**Source data 1.** The data used for plotting *Figure 5*.

which then activates its own expression in a positive feedback. This topology is also an incoherent feed-forward loop (type 1) and results in a band-pass filter output. Finally, the band-stop filter (***Figure 5F***) uses both $\beta$-estr and $\alpha$-factor as inputs, and the circuit topology presents a combination of negative feedback and an incoherent feed-forward loop (type 3). The circuit operates as a 'reversed' band-pass filter over $\alpha$-factor concentration, while $\beta$-estr is kept constant and only adds a baseline of $\alpha$-factor and auxin through the two top strains in the circuit.

The experimental data qualitatively agreed with the predictions, although we notice a time delay for the first two circuits and a shift in baseline for the latter two. The time delay is caused by under-estimation of sensor strain activation due to difficulty in isolating the sensor strain from the other strains through gating of the fluorescence data (***Figure 5C***) and possibly by an under-modeled delay of auxin synthesis (***Figure 5D***). The baseline shift is consistent with a higher concentration of auxin in the system in both cases. As a matter of fact, these two experiments (***Figure 5E F***) ended at a higher cell concentration than usual (running time of 12 and 14 hr, respectively), and it is known that yeast natively synthesizes auxin at saturation (***Rao et al., 2010***).

## Discussion

Here, we demonstrated the potential of synthetic multicellular circuits to generate a wide range of behaviors starting from simple activating or repressing individual strains. Instead of implementing complex circuits in isogenic populations, we designed simple monotonic circuits in different strains and allowed them to communicate using just two signaling molecules ($\alpha$-factor and auxin) or to affect their environment by attenuating those signals (through BAR1 and GH3.3). These simple constructs alone were capable of recapitulating many behaviors previously realized with synthetic gene circuits such as bistability, band-pass filters, pulses, and logic gates. While these behaviors were previously realized with fewer strains, it is important to point out that none of the strains in this study were designed with specific behaviors in mind other than monotonic activation/repression. Strain composition here plays the role of genetic fine-tuning, with a significantly lower experimental cost while maintaining high predictability. Strains with higher nonlinear response would simplify the topologies presented in this study but would not demonstrate the property of composition as much. Moreover, our last results on automated identification of circuit topologies that realize a target behavior (***Figure 5***) hint that the space of possible functional circuit architectures is larger than we explored.

We demonstrated through an extensive use of mathematical models, that these synthetic multi-cellular circuits are modular, easy-to-tune and extendable. Modularity is achieved through cell-cell communication that avoids cross-talk, and is demonstrated by combining input and output of our simple models. Multi-cellular circuits are tuned using different cell concentrations or positive-feedback architectures. Finally, we realized that we could tune the circuit behavior by extending or shortening the length of the signaling chain by adding or removing intermediate strains. We exploited this property to build the two time-pulses (***Figure 5***, A and B): a slower repression or activation dynamic allows for the opposite signal to operate first, generating the non-monotonic behaviors we observed. We imagine this property will gain more practical applications when the number of signaling molecules increases, which is the current major limitation in our system.

We reported that our most complex circuits display some discrepancies between simulations and experimental data, whose explanation is not always clear. It is possible that the delay between sending and receiving is not adequately modeled, or that factors that were not modeled, such as the effect of

metabolic load on cellular population distribution (*Blanchard et al., 2018*), are affecting those circuits. Future work should account for these factors at the modeling steps, for instance using distributions to describe input-output relationships *Thurley et al., 2018* or more complex models.

Finally, we leveraged the mathematical description to define an automated method to design behaviors according to performance specifications. Computationally, the method simulates all the possible strain combinations given a fixed number of nodes, and then scores them according to how well they reproduce the behavior of interest. As it is defined, the method is not easily scalable because the number of total simulations increases exponentially with the number of strains and does not account for differences in the initial strain concentrations (also defined as 'gains' in this study). However, we found no catastrophic failures in the experimental validations, such as a qualitatively different behavior, owing to the modularity of our system. Future efforts should be directed towards more efficient ways to simulate the networks, for instance by training a neural network on the current simulation sets to predict the output of interactions.

## Materials and methods
### Construction of yeast strains
Yeast transformations were carried out using a standard lithium acetate protocol used by *Gander et al., 2017*. Yeast cells were made competent by growing 50 ml cultures in rich media to log growth phase, then spinning down the cells and washing with sterile deionized water. Next, linearized DNA, salmon sperm donor DNA, 50% polyethylene glycol and 1 M LiOAc were combined with cell pellet and the mixture was heat shocked at 42° for 15 min. The cells were then spun down, supernatant was removed and they were resuspended in sterile deionized water and then plated on selective agar media. Transformations were done into *Saccharomyces cerevisiae* strain MATa W303 - 1A.

### Cytometry
Fluorescence intensity was measured with a BD Accuri C6 flow cytometer equipped with a CSampler plate adapter at room temperature using excitation wavelengths of 488 and 640 nm and an emission detection filter at 533 nm (FL1 channel). A total of 10,000 (20,000 for the bistable switch samples) events above a 400,000 FSC-H threshold (to exclude debris) were recorded for each sample using the Accuri C6 CFlow Sampler software at fast flow rate (66 µL/min, 22 $\mu$ core). Cytometry data were exported as FCS 3.0 files and processed using a custom Python script to obtain the mean FL1-A value for each data point.

### Data collection for sensor strains
Synthetic complete growth medium was used to grow the cells overnight from glycerol stock, while 300 µM IAM (Indole-3-acetamide) was added in all the experimental medium (also synthetic complete). All yeast cultures in all experiments of this study were grown in a 30° shaker incubator at 250 rpm in 14 ml Corning Falcon polypropylene round-bottom tubes (cat #: 352059) unless otherwise stated. Experiments involving time course data were taken during log phase via the following preparation: 16 hr of overnight growth in the synthetic complete medium in a shaker incubator followed by dilution to 30 events/$\mu$L into fresh, room-temperature medium. After 10 hr of growth at 30°, we performed a new dilution to 30 events/$\mu$L in 3 ml of medium, added the inducers from dilution aliquots of 100 µM stocks, and started collecting 100 µL samples without replacement for measurements periodically until the completion of the experiment.Ten thousand events were collected for each condition. Experimental replicates are intended as biological replicates of the same overnight sample (replicates conducted on the same day) or the same glycerol stock (replicates conducted on different days). Fluorescence data was labeled as outlier and discarded if the negative control differed with previous measurements to an evident amount.

### Data collection for co-culture experiments
Sample preparation was conducted as described above with each strain in a separate test tube. At the start of the experiment, each strain concentration was first measured and then strains were combined at the concentration specified by the experiment. We considered the concentration of 30 events/$\mu$L as the baseline concentration for all the experiments and all concentrations are expressed as multiples

of this reference concentration. Once the samples were added together at the desired concentrations, inducers were added and measurements were taken periodically until the completion of the experiment as described for experiments with sensor strains. For the duration of the experiment, the samples were kept in a 30° shaker at 250 rpm. Experimental replicates are intended as biological replicates of the same overnight sample (replicates conducted on the same day) or the same glycerol stock (replicates conducted on different days). Fluorescence data was labeled as outlier and discarded if the negative control differed with previous measurements to an evident amount.

### Data collection for bistable switch data

The sample preparation was conducted as described above, each strain in individual test tubes. Then, each of the five strains in the bistable switch were combined together at the concentration as outlined in the main text in a 3 ml test tube kept in a 30°C shaker at 250 rpm for the duration of the experiment. Every 40 min, we performed a $\frac{1}{3}$ dilution with fresh media with 300 μM concentration of IAM, through manual pipetting. Samples of 120 μL were collected from the sample without replacement. Alternatively, samples were collected from the dilution discard when available for the duration of the experiment approximately every 3 hr. Experimental replicates are intended as biological replicates of the same overnight sample.

### Model fitting procedure and simulations

Model parameters were estimated using Matlab fminsearch function to minimize the L2-norm of the difference between observations and simulations. For sensor strains in *Figure 1*, each parameter was estimated three times on three different experimental repeats to identify mean and standard deviation. Then, the parameters used for all the simulations in this study were estimated by fitting the average of the three measurements. Model parameters for the non-sensor strains were estimated by minimizing the L2-norm of the difference between the simulations and the average of three experimental repeats as data points. Models were simulated using the Matlab code 15s function. All models parameter values are contained in Appendix 3.

### Codes and data availability

Codes and data are available at https://github.com/Alby86/MulticellularYeast, (*Carignano, 2021* copy archived at swh:1:rev:7dc5d2f016054123df8a2bdbdd9543a06e9be63d).

## Acknowledgements

We are grateful to Cameron Cordray, Samer Halabiya, and Klavins lab technicians for technical support and to Mitchell Szeto and Alex Carr for help with the Python scripts.

## Additional information

### Funding

| Funder | Grant reference number | Author |
|---|---|---|
| Office of Naval Research | N00014-16-1-3189 | Alberto Carignano Georg Seelig Eric Klavins |
| National Science Foundation | 1807132 | Alberto Carignano Eric Klavins |

The funders had no role in study design, data collection and interpretation, or the decision to submit the work for publication.

### Author contributions

Alberto Carignano, Conceptualization, Data curation, Formal analysis, Investigation, Methodology, Validation, Visualization, Writing - original draft, Writing – review and editing; Dai Hua Chen, Cannon Mallory, Investigation; R Clay Wright, Investigation, Writing – review and editing; Georg Seelig,

Funding acquisition, Supervision, Writing – review and editing; Eric Klavins, Conceptualization, Funding acquisition, Supervision, Writing – review and editing

### Author ORCIDs
Alberto Carignano ![ORCID] http://orcid.org/0000-0003-3306-9365
Georg Seelig ![ORCID] http://orcid.org/0000-0002-3163-8782
Eric Klavins ![ORCID] http://orcid.org/0000-0002-3805-5117

### Decision letter and Author response
Decision letter https://doi.org/10.7554/eLife.74540.sa1
Author response https://doi.org/10.7554/eLife.74540.sa2

## Additional files

### Supplementary files
• Transparent reporting form

### Data availability
Figure 1 - Source Data 1, Figure 2 - Source Data 1, Figure 3 - Source Data 1, Figure 4 - Source Data 1, Figure 5 - Source Data 1 contain the numerical data used to generate the figures.

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

## Appendix 1

## Model derivation for the bistable switch

Starting from the steady state expressions with general parameters:

$$\alpha \dashv IAA : x^{ss} = x_0 + \frac{a_0}{\varphi_0 + \alpha^{n_1}} \tag{1}$$

$$\alpha \mapsto GH3 : y^{ss} = y_0 + \frac{a_1 \alpha^{n_2}}{\varphi_1 + \alpha^{n_2}} \tag{2}$$

$$IAA \dashv \alpha : z^{ss} = z_0 + \frac{a_2}{\varphi_2 + IAA^{n_3}} \tag{3}$$

$$IAA \mapsto BAR1 : w^{ss} = w_0 + \frac{a_4 IAA^{n_4}}{\varphi_4 + IAA^{n_4}} \tag{4}$$

One obtains that the steady-state quantities of signaling molecules are:

$$IAA = \frac{IAA_0 + K_1 x^{ss}}{1 + K_3 y^{ss}} \tag{5}$$

$$\alpha = \frac{\alpha_0 + K_2 z^{ss}}{1 + K_4 w^{ss}} \tag{6}$$

where $IAA_0$ and $\alpha_0$ represents exogenous concentrations added to the mix, and $K_1$, $K_2$, $K_3$, and $K_4$ are the initial concentrations of each cell type (where K = 1 is the standard value).

Substituting the above expressions (1)-(4) in (5) and (6) and re-labelling the parameters, one obtains:

$$IAA = \frac{a_1 IAA_0 + a_2 IAA_0 \alpha^{n_1} + a_3 IAA_0 \alpha^{n_2} + IAA_0 \alpha^{n_1+n_2} + K_1 \left(c_1 + c_2 \alpha^{n_1} + c_3 \alpha^{n_2} + c_4 \alpha^{n_1+n_2}\right)}{b_0 + b_1 K_3 + \alpha^{n_1}\left(c_1 + c_2 K_3\right) + \alpha^{n_2}\left(d_1 + d_2 K_3\right) + \alpha^{n_1+n_2}\left(1 + eK_3\right)} \tag{7}$$

$$\alpha = \frac{f_1 \alpha_0 + f_2 \alpha_0 IAA^{n_3} + f_3 \alpha_0 IAA^{n_4} + \alpha_0 IAA^{n_3+n_4} + K_2\left(h_1 + h_2 IAA^{n_3} + h_3 IAA^{n_4} + h_4 IAA^{n_3+n_4}\right)}{g_0 + g_1 K_4 + IAA^{n_3}\left(j_1 + j_2 K_4\right) + IAA^{n_4}\left(l_1 + l_2 K_4\right) + IAA^{n_3+n_4}\left(1 + mK_4\right)} \tag{8}$$

If one simplifies the expression above considering only the highest and lowest terms, one obtains:

$$IAA = f_{IAA}\left(\alpha\right) = \frac{a_1 IAA_0 + IAA_0 \alpha^{n_1+n_2} + K_1\left(c_1 + c_4 \alpha^{n_1+n_2}\right)}{b_0 + b_1 K_3 + \alpha^{n_1+n_2}\left(1 + eK_3\right)} \tag{9}$$

$$\alpha = f_\alpha\left(IAA\right) = \frac{f_1 \alpha_0 + \alpha_0 IAA^{n_3+n_4} + K_2\left(h_1 + h_4 IAA^{n_3+n_4}\right)}{g_0 + g_1 K_4 + IAA^{n_3+n_4}\left(1 + mK_4\right)} \tag{10}$$

Finally, if one relabels the parameters using $\widetilde{n}_1 = n_1 + n_2$ and $\widetilde{n}_2 = n_3 + n_4$ , obtains the same expression as in *Figure 3C*, Model 3.

Expressions (7) and (8) have then been fitted to simulation data of the correspondent strain combinations for different α-factor and IAA concentration to estimate the values of the parameters. Accordingly, the new exponents are: $\widetilde{n}_1 = 2.003$ and $\widetilde{n}_2 = 1.78$.

| Equation | $a_1$ | $\widetilde{n}_1$ | $c_1$ | $c_4$ | $b_0$ | $b_1$ | $e$ |
|---|---|---|---|---|---|---|---|
| IAA | 101.90 | 2.003 | 4.79e + 03 | 6.16e + 06 | 494.41 | 3.21e-12 | 17.59 |

| Equation | $f_1$ | $\widetilde{n}_2$ | $h_1$ | $h_4$ | $g_0$ | $g_1$ | $m$ |
|---|---|---|---|---|---|---|---|
| α | 428.28 | 1.78 | 1.65e + 04 | 3.28 | 1.406e + 03 | 3.59e + 04 | 233.68 |

To study the existence of two stable solutions, we solved system (9) + (10) numerically with different values of the parameters $K_i$ ($0.1 \leq K_i \leq 50$ and identified the area when multiple solutions arise. In general, bistability arise for:

1. low concentration of the IAA-repressing strain ($0.1 \leq K_1 \leq 1$) and of the BAR1-synthesis strain ($0.1 \leq K_4 \leq 1$)

2. high concentration of the α-factor-repressing strain ($10 \leq K_2 \leq 50$) and the GH3-synthesis strain ($1 \leq K_3 \leq 10$)

In the bistability region, higher $K_1$ values are paired with lower $K_4$ values in their admissible range (or vice-versa) and higher $K_2$ values are paired with lower $K_3$ values in their admissible range (or vice-versa). The simulation in *Figure 3D* in the main manuscript is generated by selecting $K_1$ =0.2 and $K_2 = 25$, while varying the values of $K_3$ between 0.1 and 30, and $K_4$ in the range between 0.1 and 1.

To simulate the evolution of α-factor and IAA concentrations over time, we extended the steady state *equations (9) and (10)* above to an ODE description by fitting the following to a time series simulation of the four strains on a 1:1:1:1 ratio:

$$\dot{IAA} = \delta_{IAA} f_{IAA}\left(\alpha\right) - \beta_{IAA} IAA \tag{11}$$

$$\dot{\alpha} = \delta_{\alpha} f_{\alpha}\left(IAA\right) - \beta_{\alpha}\alpha \tag{12}$$

where $\delta_{IAA} = \beta_{IAA} = 0.01$ and $\delta_{\alpha} = \beta_{\alpha} = 0.49$. These estimated models were used to simulate the α-factor and IAA concentrations that were then fed to the IAA repressing GFP sensor model in *Figure 1B*: the resulting simulation is shown in *Figure 3F* overlapping the experimental data.

As suggested by one of the reviewers, we considered the possibility that metabolic weight affects cell growth, which in turn could affect the stability of the equilibria. As in [3], the metabolic weight is proportional to the functions $f_{IAA}\left(\alpha\right)$ and $f_{\alpha}\left(IAA\right)$ and decreases the degradation rates $\beta_{IAA}$ and $\beta_{\alpha}$ (higher metabolic rate implies slower dilution rate) and the cell population ratios (in this context, the parameters $K_1$, $K_2$, $K_3$, and $K_4$). Assuming equal exponential growth for all the strains (parameters estimated in *Figure 3—figure supplement 3*), we obtained:

$$\widetilde{\beta_{IAA}} = \beta_{IAA}\left(1 - \rho\widetilde{f_{IAA}}\left(\alpha\right)\right) \tag{13}$$

$$\widetilde{\beta_{\alpha}} = \beta_{\alpha}\left(1 - \rho\widetilde{f_{\alpha}}\left(IAA\right)\right) \tag{14}$$

where $\widetilde{f_{IAA}}\left(\alpha\right)$ and $\widetilde{f_{\alpha}}\left(IAA\right)$ are the same functions introduced before but normalized to 1, and $0 \leq \rho \leq 1$ determines the impact of the metabolic load on the growth rates. Notice that for $\rho = 0$, we obtain back (11) and (12). The parameters $K_1$, $K_2$, $K_3$, and $K_4$ are now a function of time as follows:

$$\widetilde{K_i}\left(t\right) = K_i\, e^{-\rho\widetilde{f_{IAA}}(\alpha)t} \qquad i = 1, 3 \tag{15}$$

$$\widetilde{K_i}\left(t\right) = K_i\, e^{-\rho\widetilde{f_{\alpha}}(IAA)t} \qquad i = 2, 4 \tag{16}$$

Simulations with varying values of $\rho$ from 0 to 1 (*Figure 3—figure supplement 2*) shows that high metabolic load affects the stability of the equilibria, similarly to what is shown in [3].

## Appendix 2

### Normalization procedure for logic gate vectors

Each simulation ran 3 differential equations for each node, plus pooling parameters that represent the overall concentration of auxin and alpha-factor accounting for exogenous inputs, strain secretion, and BAR1 or GH3 synthesis, the two latter quantities being dependent on the strain selection. To automate the simulations, we represented each strain as a vector of parameters: the first 9 entries being the differential equation parameters, followed by one-hot encoding vector of size 3 to represent the sensed input ([beta-estr, alpha-factor, auxin]), and a one-hot encoding vector of size 4 to represent the secreted output ([alpha-factor, auxin, BAR1, GH3]). The last entry of the vector is a 0 for repressing strains and 1 for activating strains. Along the selected strains, we simulated the 8 sensor strains (*Figure 1C, D and E*) as system outputs.

We then grouped the simulations according to the input entries of the logic tables obtained from the 3 possible permutations of the inputs ([beta,alpha],[beta,auxin],[auxin,alpha]). These groups are shaped as a vector with 4 entries that correspond to the input states ([0 0], [1 0], [0 1], [1 1]). Within each group, we normalized each vector by their minimum value and subtracted 1 (the minimum becoming 0). This normalization scheme allowed comparison between different topologies accounting for differences in the sensor strains. At the end of this process, we had 120 vectors resulting from the 2-node networks, 560 from the 3-node networks, and 1,820 from the 4-node networks.

# Appendix 3

## All model parameters

Parameter fit values Table describing parameter estimated for Models 1 in *Figure 1*. Parameters were estimated using Matlab *fminsearch* function to minimize the L2 norm of the difference between observation and simulation. Each parameter was estimated three times on three different experimental repeats. Parameter mean and standard deviation are reported in the table against strain number and function using the ->/activation and -|/repression formalism. The parameters used for all the simulation in this study were estimated by fitting the average measurement directly, and it is presented in brackets below the mean ± std values.

| | | Parameter | | | | | | | | |
|---|---|---|---|---|---|---|---|---|---|---|
| | | $K_1$ | $\delta_1$ | $K_2$ | $\psi$ | $N$ | $\delta_2$ | $K_3$ | $\delta_3$ | $B$ |
| | 1: IAA-\|GFP | 1.32e + 06 ± 5.79e + 05 (1.76e + 06) | 3.41e + 06 ± 4.25e+05 (3.13e + 02) | 7.08e + 05 ± 2.75e + 05 (8.29e + 05) | 60.52 ± 84.57 (256.87) | 0.59 ± 0.24 (0.89) | 2.46e + 05 ± 2.51e + 04 (4.41e + 5) | 2.50e05 ±3.52e + 04 (2.12e + 05) | 5.39 ± 1.81e + 94 (7.46e + 04) | 864.63 ± 166.77 (867.98) |
| | 2: IAA-\|2xGFP | 0.65 ± 0.27 (0.69) | 326.27 ± 101.52 (416.02) | 201.66 ± 59.58 (277.36) | 2.07 ± 0.51 (1.79) | 1.01 ± 0.30 (1.011) | 10.99 ± 1.63 (11.40) | 0.0019 ± 7.45e-04 (0.0011) | 0.498 ± 0.019 (0.49) | 0.0044 ± 0.0023 (0.0049) |
| | 3: IAA->GFP | 9.64e + 04 ± 2.39e + 04 (8.95e + 04) | 0.092 ± 0.086 (0.082) | 2.53e + 07 ± 8.37e + 06 (1.73e + 07) | 1.097e + 07 ± 1.835e + 06 (1.24e + 07) | 0.83 ± 0.0059 (0.836) | 1.66e + 07 ± 2.50e + 06 (1.88e + 07) | 2.66e + 04 ± 4.95e + 03 (3.15e + 04) | 2.25e + 06 ± 4.47e + 05 (1.66e + 06) | 1.98e + 04 ± 5.49e + 03 (1.46e + 04) |
| | 4: alpha-\|GFP | 5.73 ± 2.35 (6.05) | 51.20 ± 18.34 (59.54) | 4.46 ± 4.15 (10.61) | 0.72 ± 0.17 (0.67) | 0.80 ± 0.0015 (0.80) | 0.79 ± 0.23 (0.82) | 0.0026 ± 0.0028 (3.66e-04) | 1.89 ± 1.59 (1.10) | 0.0054 ± 0.0045 (0.0032) |
| | 5: alpha->Venus | 7.49e + 05 ± 1.23e + 06 (1.06e + 03) | 3.19e + 05 ± 6.36e + 05 (0.245) | 3.06e + 06 ± 3.6e + 06 (3.47e + 05) | 8.54e + 05 ± 4.96e + 05 (7.08e + 05) | 1.60 ± 0.88 (1.04) | 1.77e + 06 ± 2.58e + 06 (1.56e + 05) | 8.036e + 04 ± 1.23e + 05 (1.62e + 04) | 8.20e + 06 ± 1.19e + 07 (2.15e + 06) | 2.36e + 04 ± 3.46e + 04 (5.96e + 03) |
| | 6: beta->GFP | 48.74 ± 10.42 (50.66) | 35.04 ± 10.09 (25.86) | 8.77 ± 2.41 (11.00) | 38.14 ± 12.70 (57.12) | 1.26 ± 0.031 (1.26) | 87.98 ± 16.44 (110.43) | 0.093 ± 0.02 (0.089) | 0.16 ± 0.0035 (0.16) | 2.13e-04 ± 3.68e-05 (2.155e-04) |
| Strain # | 7: beta-\|GFP | 1.18 ± 0.6 (0.89) | 0.18 ± 0.13 (0.17) | 243.78 ± 62.08 (174.32) | 9.16 ± 1.93 (6.65) | 2.12 ± 0.26 (2.18) | 0.58 ± 0.097 (0.56) | 1.82e-04 ± 7.53e-05 (1.5e-04) | 0.63 ± 0.073 (0.55) | 6.32e-04 ± 1.97e-04 (3.77e-04) |
| | Units | Molecule NucVol$^{-1}$nM$^{-1}$hour$^{-1}$ | hour$^{-1}$ | Molecule NucVol$^{-1}$ hour$^{-1}$ | Molecule NucVol$^{-1}$ | dimensionless | NucVol Molecule$^{-1}$ hour$^{-1}$ | Fluorescence Arbitrary Units (FAU) hour$^{-1}$ | NucVol Molecule$^{-1}$ hour$^{-1}$ | FAU Molecule NucVol$^{-1}$ hour$^{-1}$ |
| | Description | Ligand binding affinity of membrane protein(α)/ co-factor (IAA)/ TF (β) | Dissociation/ degradation/ dilution rate of ligand-binder complex | Maximum/ minimum transcription rate of TF activator/ repressor | Ligand-binder complex concentration producing half occupation | Hill-coefficient | Dissociation rate of DNA-TF | Fluorescence protein accumulation over time normalized to cell size | Fluorescence protein degradation/ dilution rate over time | Baseline fluorescence protein accumulation over time normalized to cell size |

Table presenting the parameters estimated for Model 1 for Single-Input/Single-Output strains in *Figures 2 and 3*. In the first column, the strain number and function is reported; the second column reports the sensor strain architecture that was used as baseline but with different output pathway; the parameters here reported refers uniquely to the output pathway: the remaining parameters are identical to the ones estimated for the baseline strain. Each parameter was fitted to data representing the average of three experimental repeats.

| | Parameter | | |
|---|---|---|---|
| Baseline strain # | $K_3$ | $\delta_3$ | $b$ |

*Continued on next page*

*Continued*

| | | Parameter | | |
|---|---|---|---|---|
| 10: alpha->IAA | 5 | 566.24 | 0.575 | 55.83 |
| 11: beta->BAR1 | 6 | 26.038 | 1.92e-06 | 0.35 |
| 12: beta->IAA | 6 | 3.48e + 03 | 0.16 | 0.21 |
| 13: beta->alpha | 6 | 121.60 | 0.062 | 0.14 |
| 14: IAA->alpha | 3 | 2.285 | 0.28 | 0.74 |
| 15: beta->GH3 | 6 | 109.96 | 36.71 | 1.60e-04 |
| 16: IAA->BAR1 | 3 | 1.89 | 1.83e-13 | 0.365 |
| 17: alpha->GH3 | 5 | 582.32 | 368.67 | 1.17e-14 |
| 18: alpha-\|IAA | 4 | 4.09 | 8.74e-11 | 47.78 |
| 19: beta-\|IAA | 7 | 2.024e + 11 | 4.29e + 10 | 1.85e + 09 |
| 20: beta-\|alpha | 7 | 0.077 | 1.77 | 0.18 |
| 21: IAA-\|alpha | 1 | 471.73 | 0.41 | 1.34 |
| 22: alpha->alpha | 5 | 419.52 | 2.10e + 04 | 2.32e + 04 |
| 23: IAA->IAA | 3 | 775.05 | 0.84 | 780.90 |
| 24: alpha-\|BAR1 | 4 | 0.0014 | 3.74e + 07 | 1.096e + 08 |
| Strain # | 25: IAA-\|GH3 | 1 | 225.51 | 42.46 | 3.14e-06 |
| | Units | nM hour$^{-1}$ | NucVol Molecule$^{-1}$ hour$^{-1}$ | nM Molecule NucVol$^{-1}$ hour$^{-1}$ |
| | Description | Output protein accumulation over time | Output protein degradation/dilution rate | Baseline output protein accumulation over time |

Table reporting parameter values for the two Multi-Input/Single-Output (MISO) strains in *Figure 1E*. The parameters were estimated by minimizing the L2 norm of the difference between the simulation of Model 2 (*Figure 1*) and the average of three experimental repeats.

| | | Strain # | | | |
|---|---|---|---|---|---|
| | | 8: IAA->GFP\|-alpha | 9: alpha->GFP\|-IAA | Units | Description |
| | $K_1$ | 0.0036 | 3.14E-06 | Molecule NucVol$^{-1}$nM$^{-1}$ hour$^{-1}$ | Ligand binding affinity of membrane protein($\alpha$)/co-factor (IAA)/ TF ($\beta$) |
| | $\delta_1$ | 0.0021 | 6.74E-04 | hour$^{-1}$ | Dissociation/degradation/dilution rate of ligand-binder complex |
| | $K_2$ | 6.92E-05 | 1.34E-19 | Molecule NucVol$^{-1}$nM$^{-1}$hour$^{-1}$ | Ligand binding affinity of membrane protein($\alpha$)/co-factor (IAA)/ TF ($\beta$) |
| | $\delta_2$ | 0.0047 | 7.46E-05 | hour$^{-1}$ | Dissociation/degradation/dilution rate of ligand-binder complex |
| | $K_3$ | 0.059 | 298.86 | Molecule NucVol$^{-1}$nM$^{-1}$hour$^{-1}$ | Maximum/minimum transcription rate of TF activator/repressor |
| | $\phi_1$ | 0.3 | 1.3 | Molecule NucVol$^{-1}$ | Ligand-binder complex concentration producing half occupation |
| | $n_1$ | 0.31 | 0.003 | dimensionless | Hill coefficient |
| | $\delta_3$ | 3.235 | 574.73 | NucVol Molecule$^{-1}$ hour$^{-1}$ | Dissociation rate of DNA-TF |
| | $K_4$ | 74 | 0.012 | Molecule NucVol$^{-1}$nM$^{-1}$hour$^{-1}$ | Maximum/minimum transcription rate of TF activator/repressor |
| | $\phi_2$ | 1.93 | 0.13 | Molecule NucVol$^{-1}$ | Ligand-binder complex concentration producing half occupation |
| | $n_2$ | 0.85 | 0.6 | dimensionless | Hill coefficient |
| | $\delta_4$ | 58.34 | 0.13 | NucVol Molecule$^{-1}$ hour$^{-1}$ | Dissociation rate of DNA-TF |
| | $K_5$ | 0.015 | 0.015 | FAU hour$^{-1}$ NucVol Molecule$^{-1}$ | Background fluorescence protein accumulation over time normalized to cell size |
| | $\phi_1$ | 0.32 | 0.043 | FAU hour$^{-1}$ | Fluorescence protein synthesis by the activator over time normalized to cell size |
| | $\phi_2$ | 0.87 | 2.09E + 03 | dimensionless | Repressor molecule number needed to degrade one activating molecule |
| Parameter | $\delta_5$ | 0.9 | 0.52 | hour$^{-1}$ | Fluorescence protein degradation/dilution rate over time |

