## [Editor Report]

This paper used multiple strains to build gene circuits and demonstrate the modular composition of strain circuits with an automated design strategy to achieve a target behavior from a large space of possible functional circuit architectures. This paper provides synthetic biologists with an alternative solution for the problems of scalability, robustness, and modularity.

---

## [Decision Letter]

**Decision letter after peer review:**

Thank you for submitting your article "Modular, robust and extendible multicellular circuit design in yeast" for consideration by *eLife*. Your article has been reviewed by 2 peer reviewers, one of whom is a member of our Board of Reviewing Editors, and the evaluation has been overseen Naama Barkai as the Senior Editor. The reviewers have opted to remain anonymous.

Essential revisions:

1. One of the major concerns is that I am not sure if and how the authors consider the cell growth and growth feedback into their design and modeling? The growth rates could heavily depend on the burden caused by the circuits, and host cell growth could dilute the gene expression. Will a small difference in growth rates lead to the imbalanced ratio of the strain or even loss of one of the strains in the long term? Especially in the experiment with the concentration of the upstream strain increased to10-fold. If appliable, the authors could discuss the relevance of growth rate and growth feedback in the design of multistrain circuits when they talk about the discrepancies between the simulations and experimental data. Here are some relevant publications (PMID: 29414718, 32251409)

2. The multicellular circuits in the currents look much more complicated than the original design. For example, the original toggle switch only needs two genes. In contrast, now four strains are needed in the authors' multistrain toggle switch. Similar to the logic gates. So it will add complexity and limits the scalability and robustness. What we really need is the simplicity and modularity of the circuit design is still maintained for the multistrain circuit. An in-depth discussion is expected in the revised manuscript.

3. Looks like the increased nonlinearity with external positive feedback was not used at all for the multistrain circuit designs. If so, what is the purpose of this section? Would that be used for the bistable switch? Maybe two strains instead of four is enough to make a multistrain bistable switch.

4. In the introduction, I suggest the author discuss the potential loss of modularity in the single-strain circuit but which can be solved with multiple strains. Some relevant publications (PMID: 33558556).

5. While the modeling framework can generally be applicable to all the strains, it would be better for the audience to understand if the authors specify what x1 and χ2 represent in the circuits.

6. What is the potential mechanism of ultrasensitivity in the strain in Figure 1D 2nd panel where n=2.2 was found? Would using the 1st-order Hill repressing function add or decrease some of the nonlinearity if you compare activation cases where a linear function is used?

All the reference of figures needs 'Figure ' or 'Figure' before figure number/panel

Figure 2A, what is the yellow semicircle? Should it be removed?

What is the reason why there are only 3 data points in Figure 2D-panel 4?

How did the author decide which data points are used for fitting while the rest for validation?

7. The area of bistability in Figure 3D needs to be defined.

Page 9, typo, the toggle switch circuit should be four-strain, not five-strain.

Add unit for the growth rate in Figure S10.

8. Orthogonality or no cross-talk between signaling molecules should be supported by either references or the first-hand data in this work. Please add this information when combing signals was mentioned at the first time in line 130. This data could further support the claim in line 406 that modularity is achieved through cell-cell communication that avoids cross-talk.

9. Line 309, based on the estimated growth rates in SI Figure 10, growth rates are actually statistically different by using t-test. Why did the authors say "we could not detect any growth difference…" here?

10. Line 389, what do the authors mean by "cytometrically isolating the sensor strain from other strains"? Do you mean data gating or cell sorting? The cytometer Accuri C6 cannot do cell sorting.

11. Line 418, it is unclear what individual cell difference that the authors refer to? Please be explicit about the difference.

12. Line 426, similarly, it is unclear what differences are in the initial strain populations? Please be explicit about the difference.

13. Consider changing the chemical formula H2O (lines 435 and 438) to sterile deionized water if applicable.

14. Line 439, please specify as yeast *Saccharomyces cerevisiae* strain MATa W303-1A.

15. Line 441, when using CSampler plate adapter, what is the temperature of the plate exposed to during the measurement?

16. For others to reproduce the results, please provide the details.

17. Line 444, what is the flow rate of cytometer?

18. Line 451, in 16 hrs of overnight growth, what is the rpm setting of the shaker? Are the shaker settings the same across all the experiments? What is the vessel, a tube or a plate, and what catalog number from which company?

19. Line 452, in 10 hrs of growth, what is the vessel? Is it the same rpm setting? After dilution to 3 mL medium, what is the vessel, a tube or a plate?

20. Line 453, what is the stock concentration of the inducers? 1000x stock?

21. Line454, when taking 100 uL samples periodically, do the author replenish 100 uL fresh medium after each sampling?

22. Line478, did the authors replenish fresh medium after sampling 120 uL every 3 hours?

23. Line 33. This sentence "Albeit very successful, this approach shows its limitations when it comes to scalability and robustness." is redundant.

All the numbering of the figures in the main text did not start with "Figure." Please add it.

24. Line 123, *A. thaliana* and *C. papaya* should be italic.

25. Line 155, please indicate where readers could find the data of the auxin activating strains which have a 3 fold-change activation.

26. The labeling of figure panels which use uppercase A,B,C… is not consistent with the labeling in the caption which use lowercase a, b, c….

27. SI Figure 8 and SI Figure 11 are better re-named as SI Note 1 and SI Note 2 because they are not figures.

28. Line 292, should "a set concentrations" be written as "a set of concentrations"?

29. Line 296, should Figure 3E be Figure 3F?

30. It is much clear to draw the bistable switch and the fifth reporter strain in Figure 3F.

---

## [Author Response]

Essential revisions:1. One of the major concerns is that I am not sure if and how the authors consider the cell growth and growth feedback into their design and modeling? The growth rates could heavily depend on the burden caused by the circuits, and host cell growth could dilute the gene expression. Will a small difference in growth rates lead to the imbalanced ratio of the strain or even loss of one of the strains in the long term? Especially in the experiment with the concentration of the upstream strain increased to10-fold. If appliable, the authors could discuss the relevance of growth rate and growth feedback in the design of multistrain circuits when they talk about the discrepancies between the simulations and experimental data. Here are some relevant publications (PMID: 29414718, 32251409)

We thank the reviewer for their support and detailed feedback.

We added references to the publications suggested by the author. Furthermore, we expanded our analysis to address the effect of growth feedback into our design.

We believe growth feedback is most relevant in the context of the bistable switch, since it is our most complex circuit and involves feedback regulation. We incorporated growth feedback in our original model as proposed in [Sadeghpour et al., 2017; DOI: 10.1007/s40484-017-0100-y]. Simulations with varying metabolic weight show that the system can indeed become unstable and could provide a useful explanation for some discrepancies between the experimental data and ideal model that we previously reported. We added these new results to the main text and as Figure 3—figure supplement 2.

We edited the paper as follows (line 329-338):

“An alternative explanation is that metabolic load affects the growth rates depending on strain circuit activity (Sadeghpour et al. (2017); Zhang et al. (2020)). Accordingly, active strains grow more slowly than inactive ones, affecting their intended concentrations over time, which could result in leaving the bistability region. To test this hypothesis, we introduced metabolic load to our models (Appendix 1). Following Sadeghpour et al. (2017), we assumed metabolic load to be linearly proportional to (normalized) gene expression. This dependency is captured by a parameter, where represents no impact and represents a high impact on growth rate (no growth at maximum gene expression). Simulations show that a behavior0 ≤ ρ ≤ 1 ρ = 0 ρ = 1 qualitatively similar to the loss of stability after 25 hours is possible for high metabolic load (supplement 2) according to this model representation. ρ ≥ 0. 5 ), although at a later time than the experimental data (~36hrs, Figure 3-figure. Moreover, since growth rates are statistically identical for active and inactive strains (Figure 3—figure supplement 3), we concluded that metabolic load is unlikely to significantly affect our bistable circuit. This result is consistent with previous work supporting robustness of the repressive circuit topology to growth feedback (Zhang et al. (2020)).”

Also, line 454-456 in the conclusions now reads:

“It is possible that the delay between sending and receiving is not adequately modeled, or that factors that were not modeled, such as the effect of metabolic load on cellular population distribution Blanchard et al. (2018), are affecting those circuits.”

2. The multicellular circuits in the currents look much more complicated than the original design. For example, the original toggle switch only needs two genes. In contrast, now four strains are needed in the authors' multistrain toggle switch. Similar to the logic gates. So it will add complexity and limits the scalability and robustness. What we really need is the simplicity and modularity of the circuit design is still maintained for the multistrain circuit. An in-depth discussion is expected in the revised manuscript.

We thank the reviewers for bringing up the important point of scalability and robustness. The goal of our multicellular approach was to prove that a set of strains that individually can only monotonically activate or repress gene expression are sufficient to reproduce complex behaviors through composition. In the example of the bistable switch, the selected strains were not individually designed with that behavior in mind: higher Hill coefficient would be an obvious improvement for instance and could be fairly easily achieved through strain engineering. However, thanks to the composable nature of multicellular circuits, we still are able to demonstrate bistability without the need to engineer any tailor-made gene circuit.

Our goal is to show that distributed computation can be an alternative to genetic fine tuning. ‘Multicellular tuning’ has the advantage of being experimentally simpler and cheaper, while maintaining high predictive power. We incorporated these important points to the result section and the discussion.

We edited the paper as follows (line 292-295):

“This result is not unexpected given the low Hill coefficients of the two repressive circuits (0.8 and 1 respectively). Rather than re-designing the gene regulatory circuits internal to these strains to be more suitable for a bistable switch, we decided to take advantage of the composable nature of the system and incorporate more strains to the architecture.”

And in the conclusion (line 433-439):

“While these behaviors were previously realized with fewer strains, it is important to point out that none of the strains in this study were designed with specific behaviors in mind other than monotonic activation/repression. Strain composition here plays the role of genetic fine-tuning, with a significantly lower experimental cost while maintaining high predictability. Strains with higher nonlinear response would simplify the topologies presented in this study but would not demonstrate the property of composition as much.”

3. Looks like the increased nonlinearity with external positive feedback was not used at all for the multistrain circuit designs. If so, what is the purpose of this section? Would that be used for the bistable switch? Maybe two strains instead of four is enough to make a multistrain bistable switch.

Although those strains were not directly used for the multistrain circuits, they are a proof of concept that strain composition can tune the level of nonlinearity of the response. This concept is the cornerstone of the bistable switch topology. As the reviewer points out, it is likely that, by using these positive feedback strains, we could generate a bistable system. However, while positive feedback can switch between and OFF state to an ON state, it cannot, by itself, switch back. Alternatively, one can genetically incorporate a repressive element to the external positive feedback strains, and it is possible that this strategy would reduce the number of strains in the bistable switch to two. However, we wanted to demonstrate that strain composition achieves the same results as genetically refining the strains.

4. In the introduction, I suggest the author discuss the potential loss of modularity in the single-strain circuit but which can be solved with multiple strains. Some relevant publications (PMID: 33558556).

We thank the reviewers for pointing out this reference. We have added it to the introduction (line 37-38):

“Furthermore, competition over shared gene expression resources makes unrelated circuits unintentionally coupled Zhang et al. (2021).”

5. While the modeling framework can generally be applicable to all the strains, it would be better for the audience to understand if the authors specify what x1 and χ2 represent in the circuits.

We thank the reviewers for pointing out this lack of clarity in our presentation. We have added this clarification to the text in line 179-181:

“For instance, for the auxin sensor in Figure 1B, x1 represents the TIR1-auxin complex concentration, χ2, the dCas9-VP64-auxin degron population and x3, GFP concentration.”

6. What is the potential mechanism of ultrasensitivity in the strain in Figure 1D 2nd panel where n=2.2 was found? Would using the 1st-order Hill repressing function add or decrease some of the nonlinearity if you compare activation cases where a linear function is used?

We believe that this ultrasensitivity is caused by the chromatin remodeling effect of Mxi1, which is genetically fused to dCas9 in our circuit, as suggested by [Gander et al., 2017]. The value of n=2.2 is consistent with what was estimated in that paper using a 1st-order Hill repressing function, suggesting that ultrasensitivity is not an artifact of our modeling strategy. In the manuscript, we edited lines 195-196:

“(dCas9-Mxi1 confers stronger and more consistent repression than dCas9 alone, leading to ultrasensitivity, Gander et al. (2017))”.

All the reference of figures needs 'Figure ' or 'Figure' before figure number/panel.

We thank the reviewers for pointing out this inconsistency in the text. We have edited the text accordingly.

Figure 2A, what is the yellow semicircle? Should it be removed?

We thank the reviewers for pointing out this lack of clarity in our presentation. We edited the caption of the figure to clarify the purpose of the yellow semicircle. The yellow semicircle represents the previous output (fluorescence), now swapped for auxin synthesis (blue semicircle). The strains are genetically identical with the exception of the output gene swap. We edited the caption of Figure 2A. It now says:

“To model multi-strain cascades, models for individual strains are concatenated and only the last differential equation of the first model is fit: the GFP output (dashed yellow semi-circle) is substituted with IAA (solid blue semi-circle).”

What is the reason why there are only 3 data points in Figure 2D-panel 4?

We thank the reviewers for pointing out this lack of clarity in our presentation. All strain datasets collected for Figure 2 comprise of at least 8 data points per strain. In Figure 2D-panel4, there are actually 3 data points for each of the 4 IAA concentrations, which makes for 12 data points. Hence, that dataset is in line with the other strains. The concentrations were selected to cover the operational range of the β-estr activating strains and IAA repressing strains.

How did the author decide which data points are used for fitting while the rest for validation?

We thank the reviewers for pointing this out. The two datasets come from two different experiments. We added a reference to the main text in lines 231-233:

“…we collected two datasets on two different experiments composed of four data points each: we used one for fitting (yellow dots, first experiment), and the other for validation (orange dots, second experiment, Figure 2A)…”

7. The area of bistability in Figure 3D needs to be defined.

We thank the reviewers for pointing this out. We have added a description of the bistability region in Appendix 1, second page, starting from the first paragraph.

Page 9, typo, the toggle switch circuit should be four-strain, not five-strain.

We thank the reviewers for pointing this out. We have added a description of the bistability region in Appendix 1, second page, starting from the first paragraph.

Add unit for the growth rate in Figure S10.

We thank the reviewers for pointing this out. We have edited the text (added *min^{-1}*) of Figure S10 accordingly.

8. Orthogonality or no cross-talk between signaling molecules should be supported by either references or the first-hand data in this work. Please add this information when combing signals was mentioned at the first time in line 130. This data could further support the claim in line 406 that modularity is achieved through cell-cell communication that avoids cross-talk.

We thank the reviewers for bringing up the important point of signal orthogonality. We added Figure 1—figure supplement 7 that shows a high degree of orthogonality between the 3 different signals across the repressive sensors. The signals were combined and compared to the individual signals and little difference was detected. The figure is also referenced in the main text in lines 198-202 as:

“We tested signal orthogonality by adding pairwise combinations of inducers at saturation concentration to single-input repressive sensors (Figure 1—figure supplement 7). Variations across the treatments are contained within 11% of the nominal value for signal concentrations within the range used for circuit experiments.

A maximum increase of 24% above the baseline is observed for the IAA-sensor in a regime where both α factor and auxin are at or above saturating concentrations (1uM α-factor and 10uM IAA). However, this combination is not used in any circuit experiment.”

9. Line 309, based on the estimated growth rates in SI Figure 10, growth rates are actually statistically different by using t-test. Why did the authors say "we could not detect any growth difference…" here?

We thank the reviewers for pointing out the lack of statistical validation.

We performed and added to Figure 3—figure supplement 3 an ANOVA test that does not find the growth rates to be statistically different across strains with a 90% confidence. Pairwise t-tests also show no growth difference between the strains that sense-and-secrete signaling molecules, but did find differences if comparing with the sensor strain with 95% confidence. However, changes in the sensor strain concentration do not affect the existence of two stable equilibria in the system. According to our simulations, the dilution scheme of the bistable switch does not result in the sensor strain taking over the co-culture during the duration of the experiment since the difference is minimal. Furthermore, it is not obvious that statistical tests would be meaningful with a low sample size (n=3) as in this case. We added this analysis to Figure 3—figure supplement 3 and referenced it in the main text, line 328: “…we could not detect any growth difference between the single-input/single-output strains using an ANOVA test, Figure 3—figure supplement 3.”

10. Line 389, what do the authors mean by "cytometrically isolating the sensor strain from other strains"? Do you mean data gating or cell sorting? The cytometer Accuri C6 cannot do cell sorting.

We thank the reviewers for pointing out this lack of clarity in the text. We have edited the text, and now lines 419-420 read: “…difficulty in isolating the sensor strain from the other strains through gating of the fluorescence data.”

11. Line 418, it is unclear what individual cell difference that the authors refer to? Please be explicit about the difference.

We thank the reviewers for pointing out this lack of clarity in the text. We have edited the text, and now lines 454-455 read: “…factors that were not modeled, such as the effect of metabolic load on cellular population distribution…”.

12. Line 426, similarly, it is unclear what differences are in the initial strain populations? Please be explicit about the difference.

We thank the reviewers for pointing out this lack of clarity in the text. We have edited the text, and now lines 464-465 read: “…does not account for differences in the initial strain concentrations (also defined as ’gains’ in this study)…”.

13. Consider changing the chemical formula H2O (lines 435 and 438) to sterile deionized water if applicable.

We thank the reviewers for pointing out this lack of clarity in the text. Lines 473 and 476 (former lines 435 and 438) now read: “sterile deionized water”.

14. Line 439, please specify as yeast *Saccharomyces cerevisiae* strain MATa W303-1A.

We thank the reviewers for pointing out this lack of clarity in the text. Line 477 (former line 439) now reads: “*Saccharomyces cerevisiae* strain MATa W303-1A”.

15. Line 441, when using CSampler plate adapter, what is the temperature of the plate exposed to during the measurement?

We thank the reviewers for pointing out this lack of clarity in the text. We have added to the text at line 480: “at room temperature”.

16. For others to reproduce the results, please provide the details.

Details are included below for specific questions.

17. Line 444, what is the flow rate of cytometer?

We thank the reviewers for pointing out this lack of clarity in the text. We have added line 483: “at fast flow rate (66L/min, 22 core)”.

18. Line 451, in 16 hrs of overnight growth, what is the rpm setting of the shaker? Are the shaker settings the same across all the experiments? What is the vessel, a tube or a plate, and what catalog number from which company?

We thank the reviewers for pointing out this lack of detail in the text. We have added line 489-491: “All yeast cultures in all experiments of this study were grown in a 30C shaker incubator at 250rpm in 14 ml Corning Falcon polypropylene round-bottom tubes (cat #: 352059) unless otherwise stated.”

19. Line 452, in 10 hrs of growth, what is the vessel? Is it the same rpm setting? After dilution to 3 mL medium, what is the vessel, a tube or a plate?

We thank the reviewers for pointing out this lack of clarity in the text. We have edited the text as described in the previous point.

20. Line 453, what is the stock concentration of the inducers? 1000x stock?

We thank the reviewers for pointing out this lack of clarity in the text. We added to lines 494-495: “from dilution aliquots of 100M stocks”.

21. Line454, when taking 100 uL samples periodically, do the author replenish 100 uL fresh medium after each sampling?

We thank the reviewers for pointing out this lack of clarity in the text. We added line 495 that reads: “without replacement”.

22. Line478, did the authors replenish fresh medium after sampling 120 uL every 3 hours?

We thank the reviewers for pointing out this lack of clarity in the text. We have edited lines 519-520 as follow: “…from the sample without replacement. Alternatively, samples were collected from the dilution discard when available…”.

23. Line 33. This sentence "Albeit very successful, this approach shows its limitations when it comes to scalability and robustness." is redundant.

We thank the reviewers for pointing out this redundancy in the text. We have removed the redundant line from the text.

All the numbering of the figures in the main text did not start with "Figure." Please add it.24. Line 123, *A. thaliana* and *C. papaya* should be italic.

We thank the reviewers for pointing out this lack of clarity in the text. We have edited the text accordingly and now the numbering starts with ‘Figure’. Also, the name of the two species are now in italic.

25. Line 155, please indicate where readers could find the data of the auxin activating strains which have a 3 fold-change activation.

In the main manuscript, at line 533, we reported that all data can be retrieved on the github repository (https://github.com/Alby86/MulticellularYeast.git). The data for that strain are also available in the AllData.xlsx spreadsheet.

26. The labeling of figure panels which use uppercase A,B,C… is not consistent with the labeling in the caption which use lowercase a, b, c….

We thank the reviewers for pointing out this inconsistency in the text. We have edited the text accordingly and it is now consistently using uppercase letters.

27. SI Figure 8 and SI Figure 11 are better re-named as SI Note 1 and SI Note 2 because they are not figures.

We thank the reviewers for pointing out this inconsistency in the text. We have relabelled them as Appendix 1 and Appendix 2.

28. Line 292, should "a set concentrations" be written as "a set of concentrations"?

We thank the reviewers for pointing out this typo. Line 309 now reads: “a set of concentrations”.

29. Line 296, should Figure 3E be Figure 3F?

We thank the reviewers for pointing out this mistake. We now reference Figure 3F.

30. It is much clear to draw the bistable switch and the fifth reporter strain in Figure 3F.

We thank the reviewers for pointing out this lack of clarity in the text. We have modified Figure 3F to incorporate the reporter strain.